# The chromatin-associated *lncREST* ensures effective replication stress response by promoting the assembly of fork signaling factors

Luisa Statello [1,2] ✉, José Miguel Fernandez-Justel [1,2], Jovanna González[1,2], Marta Montes[1,2], Alessia Ranieri[1,2], Enrique Goñi [1,2], Aina M. Mas [1,2] & Maite Huarte [1,2] ✉

Besides the well-characterized protein network involved in the replication stress response, several regulatory RNAs have been shown to play a role in this critical process. However, it has remained elusive whether they act locally at the stressed forks. Here, by investigating the RNAs localizing on chromatin upon replication stress induced by hydroxyurea, we identified a set of lncRNAs upregulated in S-phase and controlled by stress transcription factors. Among them, we demonstrate that the previously uncharacterized lncRNA *lncREST* (long non-coding RNA REplication STress) is transcriptionally controlled by p53 and localizes at stressed replication forks. *LncREST*-depleted cells experience sustained replication fork progression and accumulate un-signaled DNA damage. Under replication stress, *lncREST* interacts with the protein NCL and assists in engaging its interaction with RPA. The loss of *lncREST* is associated with a reduced NCL-RPA interaction and decreased RPA on chromatin, leading to defective replication stress signaling and accumulation of mitotic defects, resulting in apoptosis and a reduction in tumorigenic potential of cancer cells. These findings uncover the function of a lncRNA in favoring the recruitment of replication proteins to sites of DNA replication.

Replication stress (RS) is a common phenomenon occurring during DNA replication and a threat to genome integrity of normal and cancer cells. It arises when the replication fork encounters obstacles or deficiencies that hinder its progression. RS is one of the major sources of DNA damage, leading to fork collapse and double strand breaks if not resolved. To prevent damage and maintain genomic stability, cells have evolved a complex network of signaling pathways that respond rapidly to replication stress[1]. The study of the mechanisms underlying this response has mainly focused on the assembly and disassembly of protein complexes, involving a highly coordinated repertoire of post-translational modifications such as phosphorylation,

sumoylation and acetylation, rapidly engaging the correct pathways to be activated in response to specific types of stress[2].

Multiple factors are known to participate, including DNA repair proteins, checkpoint kinases, replication fork stabilizers, protein chaperones and chromatin modifiers among others[3]. These factors are dynamically recruited and disassembled at the damaged forks in a finely regulated manner, allowing cells to re-enter the cell cycle undamaged. Understanding their regulation can provide important insights into the mechanisms of DNA damage response and potentially inform new strategies for treating diseases associated with DNA damage.

[1]Center for Applied Medical Research, University of Navarra, Pio XII 55 Ave, 11 31008 Pamplona, Spain. [2]Institute of Health Research of Navarra (IdiSNA), Cancer Center Clínica Universidad de Navarra (CCUN), Pamplona, Spain. ✉e-mail: lstatello@unav.es; maitehuarte@unav.es

The relocation and concentration of factors in the proximity of the damaged forks are required to effectively address the urgent cellular needs. In this context, the role of non-protein factors has received limited attention despite their implication in the formation of subcellular compartments that facilitate nuclear processes[4–6]. In particular, long non-coding RNAs (lncRNAs) have significant regulatory potential and may be critical for orchestrating rapid responses to mitigate damage. LncRNAs are noncoding RNAs recently re-defined based on their length of more than 500 nt[7]. Similar to mRNAs, most of them are transcribed by RNA polymerase II, polyadenylated, and spliced. Their distinctive feature as non-translated RNAs is frequently linked to nuclear localization, where lncRNAs have been shown to regulate gene expression and signaling pathways by interacting with DNA, RNA, or proteins. Of particular interest, nuclear-retained lncRNAs that are chromatin-associated are emerging as regulatory layers of fundamental functions such as DNA replication and maintenance of genome stability[8–12]. Importantly, they also have the capacity to impact nuclear function by promoting subnuclear compartments that concentrate and localize factors to their functional sites[4].

Here, we investigated the involvement of chromatin-associated lncRNAs in the response to replication stress in human cells, and identified *lncREST* as a crucial player of this response. Our results provide new insights into the interplay between RNA and protein components at the interface between DNA replication and repair, which may have important implications for the development of novel strategies to target replication stress from a therapeutic perspective.

## Results

### Replication stress induces a specific subset of chromatin-associated lncRNAs

Many chromatin-tethered RNAs are not targeted for degradation; instead, their nuclear accumulation is related to their functions[13]. We set to investigate chromatin-retained lncRNAs upon replication stress, hypothesizing that their expression may be triggered to functionally support the replication stress response. To that end, we treated HCT116 cells with 1 mM hydroxyurea (HU) for 8 h, a condition that causes G1 accumulation and transient phosphorylation of ATR (p-ATR), one of the first responders to fork stalling stimulated by the accumulation of ssDNA stretches in presence of HU-induced dNTP depletion[14] (Fig. 1a and Supplementary Fig. 1a, b). HU treatment also leads to phosphorylation of H2A.X (γH2A.X), and stabilization of phosphorylated p53 (p-p53), all markers of stress and DNA damage that are rapidly recovered by removing HU from cell culture medium (Fig. 1a and Supplementary Fig. 1b). We applied the *subRNA-seq* fractionation protocol to untreated or HU-treated cells[15] to extract chromatin-associated RNAs, as well as total cellular RNA as reference (Material and Methods, Fig. 1b and Supplementary Fig. 1c). The sequencing of total RNA (polyA+ and polyA-) from both fractions confirmed the enrichment in nascent unspliced transcripts in the chromatin by monitoring the percentage of exonic reads, which is notably higher in the total RNA preparations (Supplementary Fig. 1d). The differential expression analysis between the untreated and HU-treated samples identified 261 RNAs differentially expressed in the total fraction, and 354 in the chromatin fraction (Supplementary Data 1). The relatively low number of differentially expressed genes upon HU treatment confirms that the replication-stress response is primarily non-transcriptional[16,17]. Specifically, 95 differentially expressed lncRNAs were found in chromatin (57 upregulated and 38 downregulated), and 80 in the total fraction (66 upregulated and 14 downregulated) (Fig. 1c and Supplementary Fig. 1e). Despite the fact that the expression trend in the two fractions is similar, only 22% of differentially expressed lncRNAs in chromatin are also significantly dysregulated in the total fraction, indicating that our fractionation protocol allows the identification of a specific set of RNAs that are transcriptionally activated on chromatin, or that relocate to the

chromatin fraction in response to HU, where they often exert their function (Fig. 1d)[7,13].

To better characterize the nature of the chromatin transcripts induced by replication stress, we studied different features related to their transcriptional regulation. Using publicly available Repli-seq data[18], we found that the change in expression of these genes was associated with their replication timing: transcripts upregulated upon HU treatment were preferentially located in early replicating regions, while downregulated ones tended to have a later replication timing (Fig. 1e). Gene Ontology (GO) term enrichment analysis revealed that the differentially expressed transcripts were primarily involved in three distinct functions: chromatin and nucleosome assembly, nuclear division, and regulation of CDK activity (Supplementary Fig. 1f), possibly reflecting the DNA replication perturbations caused by HU treatment.

Since the replication stress response takes place in the S phase of the cell cycle, we analyzed the temporal expression patterns of the lncRNAs induced on chromatin of cells synchronized in the different phases of the cell cycle (Material and Methods). The analysis revealed an enrichment of lncRNAs with elevated expression levels during early S phase ($P = 0.003$) (Fig.1e).

Finally, we investigated the transcription factors binding to the promoters of differentially enriched chromatin-associated lncRNAs. Interestingly, *Enrichr* software[19] found that they were enriched for binding of FOXM1 (Fig. 1f), which controls the expression of genes involved in G1/S transition, replication, G2/M transition, and mitosis, and has been reported as a regulator of replication stress[20]. Moreover, more than 65% of the lncRNAs induced by replication stress were transcriptionally controlled by factors involved in the stress response (ATF3, BRD4, MYC, and TP53) (Fig. 1g).

In conclusion, our results indicate that replication stress triggers the expression of a group of chromatin-associated lncRNAs. Several of them are induced during the early-mid S phase of the cell cycle and regulated by the main pathways involved in the replication stress response. We hypothesized that they could play a role in cellular responses to replication stress.

### *lncREST* responds to replication and genotoxic stresses

One of the lncRNAs most highly induced on chromatin when replication stress was inflicted with hydroxyurea was the lncRNA *ENSG00000253878* (Figs. 1c, 2a–c, Supplementary Fig. 2a, b), which had previously been identified as a p53-regulated lncRNA in lung cancer cells[21], and which from now on will be referred to as *lncREST* (long non-coding RNA REplication STress). *LncREST* is a three-exon 709 nt-long transcript located in chromosome 8 (Fig. 2a) mainly chromatin associated in HCT116 cells (Supplementary Fig. 2a) and with low coding capacity according to CPAT and CSF scores (Supplementary Fig. 2c). *LncREST* is sense-intronic to *NDUFAF6*, and divergently transcribed 600 bp away from the protein-coding gene *TP53INP1* (Fig. 2a).

To gain deeper insight into *lncREST* regulation, we evaluated its expression in a double thymidine block experiment followed by the release of HCT116 cells into different cell cycle stages (Fig. 2d and Supplementary Fig. 2d. By parallel evaluation of specific cell cycle and DNA damage markers, we verified that the expression of *lncREST* along the time course was cell cycle specific and not influenced by possible induction of genotoxic stress (Fig. 2d).

Despite their related genomic localization, *lncREST* and *NDUFAF6* are independently regulated: *NDUFAF6* expression is not cell cycle regulated (Supplementary Fig. 2d–f). Moreover, *lncREST* host gene *NDUFAF6* is not affected by replication stress (Supplementary Fig. 2g). However, *TP53INP1* follows the expression profile of *lncREST* during the cell cycle, being induced upon different treatments (Supplementary Fig. 2d–f and h). On the other hand, *lncREST* displays several interesting features: (i) Its expression peaks in the S-phase of the cell cycle, similar to the G1/early S-phase cell cycle marker *cyclin E* (Fig. 2d), (ii) it shows robust induction on chromatin following treatment with

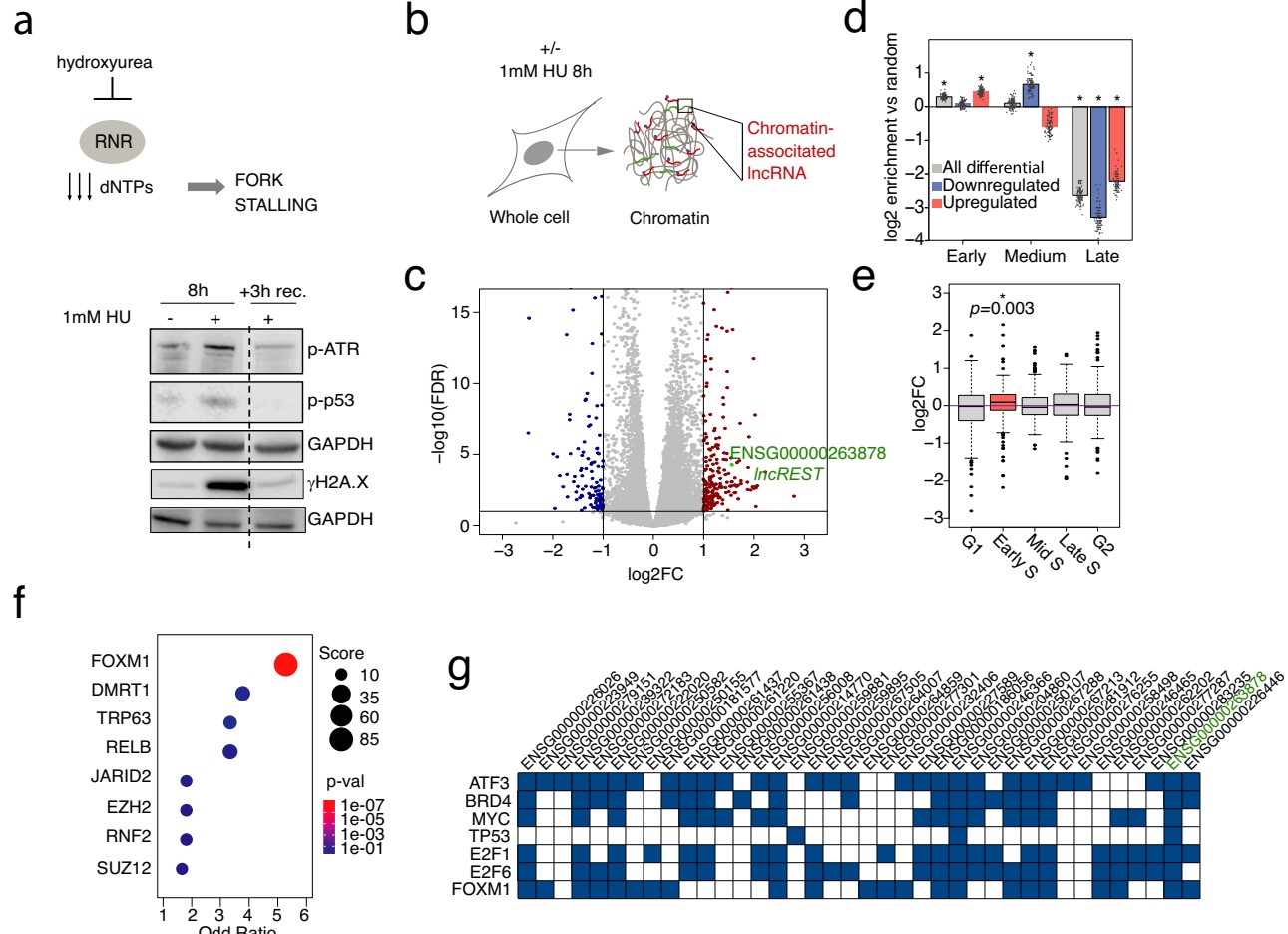

**Fig. 1 | Replication stress induces lncRNAs associated to replicating chromatin.**
**a** (Top) Mechanism of action of hydroxyurea (HU) RNR: ribonucleotide reductase. (Bottom) Immunoblot analysis HCT116 cells treated with HU 1 mM for 8 h followed by 3 h recovery shows the reversible effect of HU on replication stress markers p-ATR, p-p53 and γH2A.X. Experiments were performed twice with similar results. Source data are provided as a Source Data file. **b** Schematic of the fractionation protocol applied to isolate chromatin-associated RNAs. **c** Volcano plot showing the -log10(adjusted p-value) and the log2(fold-change) from the RNA-seq differential expression analysis, comparing the HU-treated vs untreated chromatin fractions. Transcripts with ± 1logFC (FDR < 0.05) are highlighted in blue (-1logFC) and red ( + 1logFC). DESeq2[54] two-sided, with Benjamini-Hochberg FDR correction, was used to measure differential expression. Volcano plot was drawn using ggplot2 R package. **d** Distribution of the log2(fold-change) of the HU-upregulated genes in the different cell cycle stages of a synchronized RNA-seq. Data points represent log2-ratio of the number of genes in each replication phase vs the number obtained in the 100 random sets generated. (mean ±2 x standard deviation). Source data are provided as a Source Data file **e** Temporal expression pattern analysis of the induced lncRNAs in RNA-seq data of cells synchronized in the different phases of the cell cycle. *n* = 3 (one sample wilcoxon test, no multiple testing corrections). Fold-change is defined as the ratio between the expression in one time point versus the expression in the pool of all the other points. Boxplots represent 25 to 75 percentiles, whiskers are 1.5 x interquartile range (interquartile range = percentile75-percentile25). **f** *Enrichr* analysis of the promoters of the differentially expressed genes upon HU treatment, searching against the CHEA Transcription Factor Targets database. **g** Overlap between the promoter of the upregulated lncRNAs and the binding sites of stress-related transcription factors. The blue squares on the grid show the candidate lncRNAs that have a binding site in their promoter for the given transcription factor.

HU as well as other stress agents (Fig. 2b, c, e, Supplementary Fig. 2a), (iii) it is broadly expressed in different solid tissues and cell lines and induced during RS, (Fig. 2e and Supplementary Fig. 2i–l) while its promoter is bound by transcription factors implicated in RS response (i.e.p53, BRD4, c-myc, ATF3, E2F1, E2F6 and FOXM1) (Fig. 1g and Supplementary Fig. 2m). The direct regulation of *lncREST* by the transcription factor p53 upon stress was further confirmed by p53 ChIP-seq of HCT116 cells treated with the DNA-damaging agent 5-FU[22]. This revealed a robust p53 binding peak at *lncREST* promoter region (Fig. 2f). Accordingly, *lncREST* showed significant induction by replication stress treatment in p53 wt but not p53$^{-/-}$ cells (Fig. 2g), providing further evidence of the direct role of p53 in the activation of *lncREST* expression under stress conditions.

In conclusion, our findings suggest that *lncREST* is a lncRNA that responds to replication and genotoxic stresses, and is regulated by the transcription factor p53.

**lncREST is required for the correct S-phase checkpoint signaling**
Given the general features of *lncREST* as replication stress-induced, S-phase enriched, and chromatin-associated lncRNA, we evaluated its potential role in the regulation of the replication stress response. To investigate its role in this pathway, we first knocked down *lncREST* by using two independent LNA GapmeRs (Fig. 3a and Supplementary Fig. 3a, c), and analyzed phenotypes and markers of the replication stress response. In order to make sure we are dealing with replication stress and not generally DNA damage phenotypes, since *lncREST* is also activated by agents directly causing genotoxic stress (see Fig. 2), we treated *lncREST*-depleted cells with two specific RS inducing drugs, HU and APH. When compared to control cells, *lncREST*-depleted cells showed accumulation in G2 phase upon replication stress challenge (Fig. 3b). Similar cell cycle alteration was observed in the *lncREST* knockout HCT116 clones generated using a CRISPR/Cas9 system, and when *lncREST* was silenced by CRISPRi strategy by

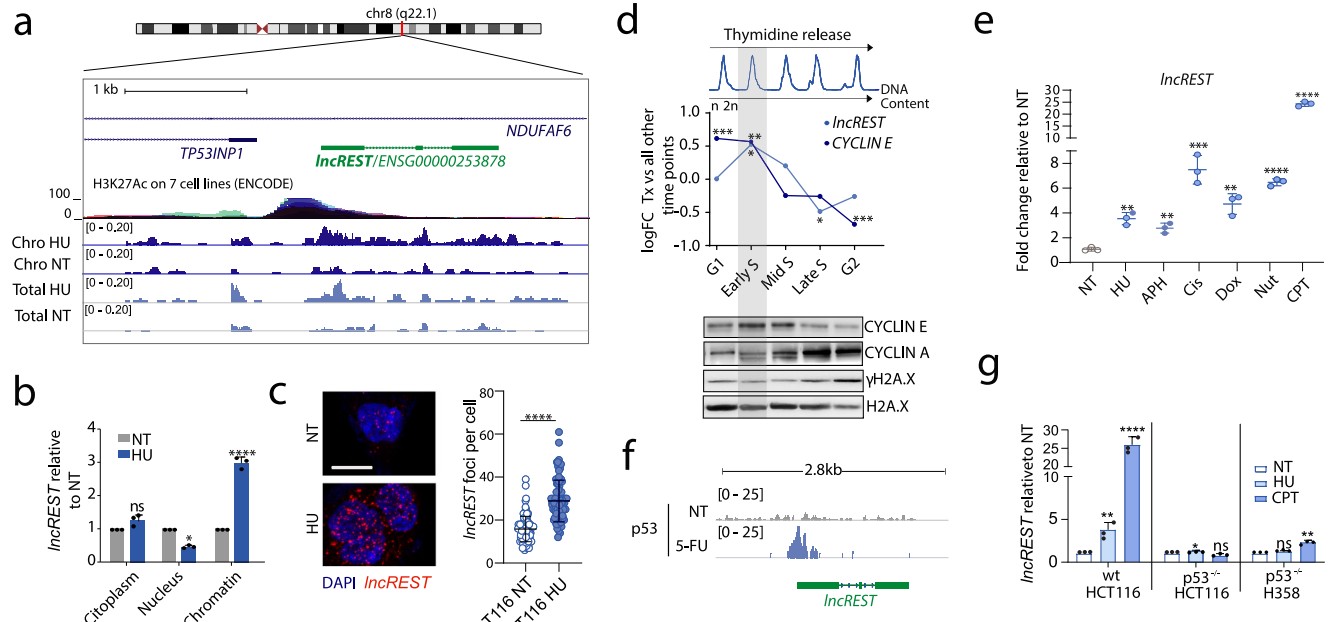

**Fig. 2 | *lncREST* is a lncRNA induced on chromatin by p53 following replication stress. a** Diagram of *lncREST* locus and RNA-seq tracks (counts per million) for chromatin and total fraction of HCT116 cells treated with 1 mM HU for 8 h. *lncREST* consists of three exons, is sense intronic to the protein coding gene *NDUFAF6*, and is located in proximity of the protein coding gene *TP53INP1*. **b** qRT-PCR analysis of *lncREST* in cellular fractions following treatment with HU. (two-tailed unpaired t-test, mean ± SD, *n* = 3 per condition) Cytoplasm *p* = 0.2441 (ns); nuclear *p* = 0.0111 (*); chromatin *p* < 0.0001 (****). **c** Representative images of *lncREST* FISH in HCT116 cells treated with PBS or HU o.n. and dot plot of *lncREST* foci quantification, mean ± STD. Three experiments were performed with similar results, at least 80 cells per condition were analyzed (*n* = 2). ****p* < 0.0001 (two-tailed unpaired t-test). Scale bar: 10 μm. Uncropped images are provided with the Source Data file. **d** From top to bottom: Flow cytometry profiles of HCT116 synchronized cells, representing the DNA content at different cell cycle stages from release; relative expression of *lncREST* and *cyclin E* measured by RNA-seq. log2FC of each time point vs all other time points is represented. *p* < 0.05, **p* < 0.01, ***p* < 0.001 (*p* = 0.017243161 for

*lncREST* in early S; *p* = 0.036741047 for lncREST in late S; for cyclin E *p* = 0.000486891, *p* = 0.00130681, *p* = 0.000137981 in G1, early S and G2, respectively; western blot of cell cycle and DNA damage markers in each time point. Source data are provided as a Source Data file. **e** qRT-PCR analysis of *lncREST* expression in HCT116 cells treated with 2 mM of hydroxyurea (HU) o.n., 40 μM aphidicolin (APH) for 24 h, 15 μM of cisplatin (cis) o.n., 5 μM of doxorubicin (Dox) for 24 hrs, 20 μM of nutlin (Nut) o.n., 10 μM of camptecin (CPT) for 8 h, relative to normal condition (NT). *n* = 3 biological replicates. ***p* < 0.01, ****p* < 0.001, *****p* < 0.0001 (mean ± SD, two-tailed unpaired t-test). **f** Integrative genomics viewer (IGV) browser snapshot of p53 ChIP-seq enrichment in control and 5-FU treated HCT116 cells. **g** qRT-PCR of *lncREST* in HCT116 wt, HCT116 p53⁻/⁻, and H358 p53⁻/⁻ cells untreated, HU and CPT treated. (mean ± SD, two-tailed unpaired t-test. *n* = 3) (HCT116 p53⁻/⁻ **p* = 0.0324 for HU treatment, H358 ***p* = 0.0026 for CPT treatment, HCT116 wt ***p* = 0.0012 for HU treatment, *****p* < 0.0001 for CPT treatment. For **b**–**e** and **g**, source data are provided as a Source Data file.

targeting a dCAS-KRAB fusion protein to the TSS of *lncREST* (Material and methods)[23] (Supplementary Fig. 3a, b, d–f, i and j), indicating the involvement of the lncRNA. Alkaline comet assay showed severe DNA damage in *lncREST* knockdown cells following HU and APH treatment compared to control cells. (Fig. 3c). The increased genomic damage in *lncREST*-depleted cells was corroborated by the presence of a higher number of micronuclei, another indicator of genomic instability, already appearing in the non-treated condition (Fig. 3d). Unexpectedly, despite the increased DNA damage, we observed a reduction of markers of replication stress and DNA-damage signaling after exposure of *lncREST* depleted cells to HU (Fig. 3e). We confirmed these results by immunofluorescence, observing a reduction of γH2A.X foci, especially when *lncREST* was knocked down in cells challenged with HU (Fig. 3f). Furthermore, while ATR was already strongly phosphorylated (p-ATR) at early time points after HU treatment in *lncREST*-proficient cells, *lncREST* knockdown cells failed to achieve or maintain the same level of ATR activation over time (Fig. 3e, g). The same accumulation of genotoxic stress associated to a reduction of γH2A.X foci and increased DNA damage shown by alkaline comet assay was observed in *lncREST* KO and dCas9/KRAB cells (Supplementary Fig. 3o–s). We concluded that cells require *lncREST* to sustain functional signaling upon DNA damage inflicted by replication stress.

Since lncRNAs have often been found to act *in cis*[24], we wanted to determine whether this phenotype could be attributed to the

regulation of *lncREST* neighbor gene *TP53INP1*, a direct activator of autophagy-dependent apoptosis with tumor suppressor activity[25,26]. Interestingly, orthogonal approaches leading to *lncREST* depletion had opposed effects on *TP53INP1* expression: while LNA-mediated *lncREST* depletion led to induction of *TP53INP1* (Supplementary Fig. 3a and d), CRISPR approaches aimed to knockdown or silence *lncREST* (i.e. CRISPR KO and dCas9/KRAB) (Supplementary Fig. 3a) led to a reduction of *TP53INP1* (Supplementary Fig. 3e and f). The latter was not entirely unexpected, given that the two genes have a head-to-head configuration sharing regulatory regions, likely affected by alteration of the chromatin state by dCas9/KRAB, or the deletion of DNA elements with CRISPR/Cas9 (Supplementary Fig. 3a). To further investigate the possible mutual regulation of *lncREST* and *TP53INP1* genes, we performed the individual KD of *lncREST*, of *TP53INP1*, or the double KD. As previously observed, the KD of *lncREST* leads to upregulation of *TP53INP1*, however *TP53INP1* KD leads to downregulation of *lncREST* (Supplementary Fig. 3g, h), and increased number of cells in G2 phase (Supplementary Fig. 3k).

To evaluate whether the checkpoint phenotypes could be mediated by the induction of *TP53INP1*, we overexpressed *TP53INP1* from a plasmid, which leads to the downregulation of *lncREST* (Supplementary Fig. 3l). Under these experimental conditions there was an increase in the number of cells in G2 phase (Supplementary Fig. 3m), in line with the previously observed for *lncREST* reduction (Fig. 3b). Upon *TP53INP1* overexpression and CTP or HU treatments we also observed

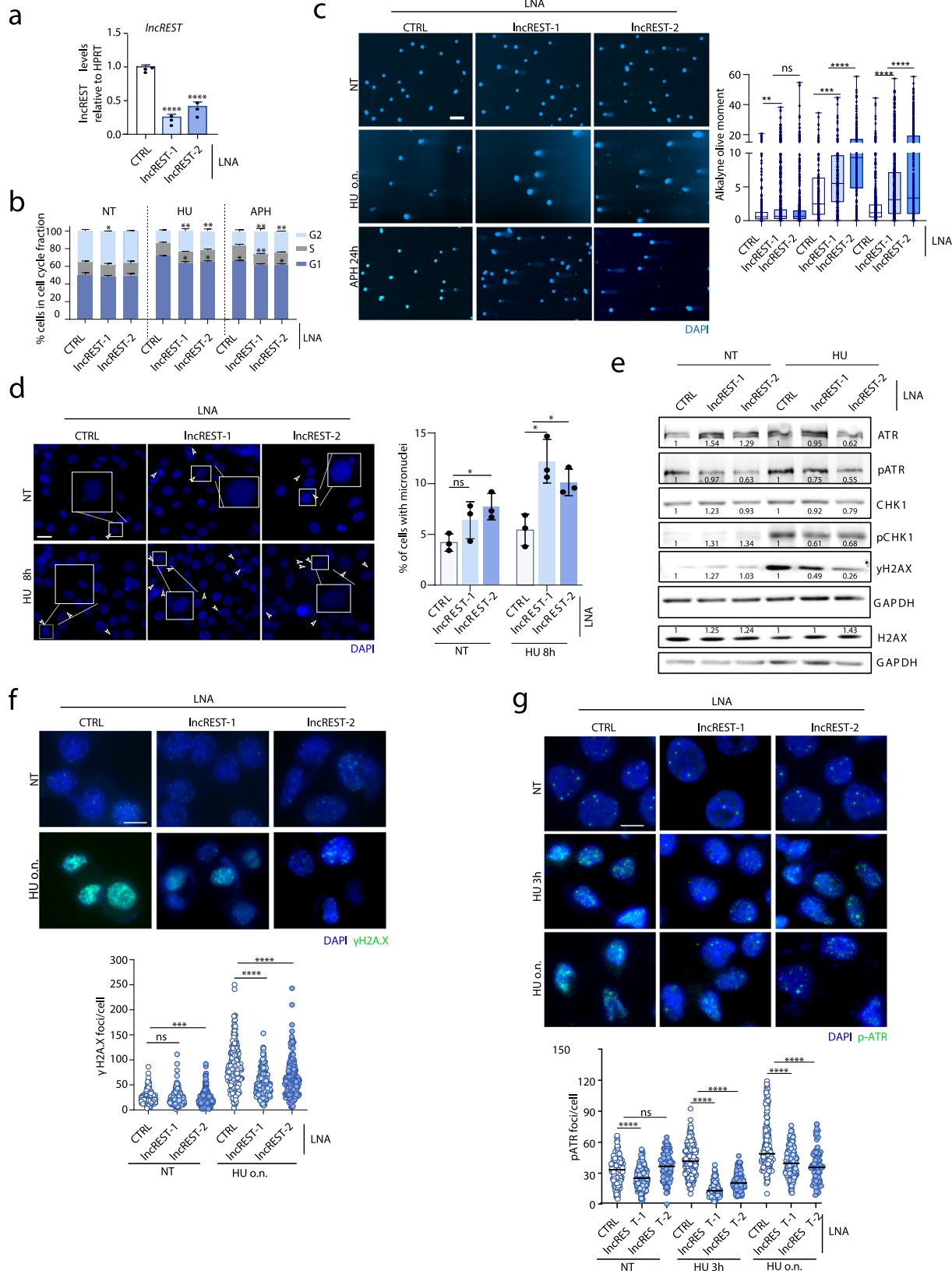

some reduction of the checkpoint marker p-ATR (Supplementary Fig. 3n), similar to *lncREST* KD. Interestingly, TP53INP1 protein is very lowly expressed, with undetectable endogenous levels (Supplementary Fig. 3n), which argues against a significant role of the protein in the experimental conditions of or study. Moreover, despite the diverse and opposed effects on *TP53INP1*, the reduction of *lncREST* expression always led to reproducible checkpoint defects (Supplementary Fig. 3o–s). This observation agrees with the notion that, while both genes are co-regulated due to their linked genomic configuration, the role of *lncREST* on the replication stress response is independent of *TP53INP1*. We therefore set to investigate the autonomous function of *lncREST* in more detail.

**Fig. 3 | *lncREST* regulates replication stress response. a** qRT-PCR of *lncREST* in HCT116 cells following depletion with two independent LNA GapmeRs. ****$p < 0.0001$ (mean ± SD, two-tailed unpaired t-test. $n = 3$). **b** Percentage of cells in each cell cycle fraction measured by flow cytometry in cells treated with 1 mM HU o.n and 40 μM aphidicolin (APH) 24 h. *$p < 0.05$, **$p < 0.01$ (mean ± SD, two-tailed unpaired t-test. $n = 3$). **c** Alkaline comet assay showing the increase ss-and dsDNA breaks in *lncREST* KD cells in NT condition and challenged with 1 mM HU o.n. and 40 μM aphidicolin (APH) 24 h. Left panel, representative images. Scale bar, 100 μm. Right panel, boxplots showing the alkaline olive moment. Whiskers are set to min/max value, boxes extend from the 25th to 75th percentiles, median is shown. At least 200 tails were analyzed for each condition ($n = 3$). **$p = 0.0081$ for CTRL KD vs *lncREST* KD1 NT, ***$p = 0.0004$ for CTRL vs *lncREST* KD1 HU, ****$p < 0.0001$ (two-tailed unpaired t-test). **d** Micronuclei in *lncREST* KD cells. Left panel, representative images. Right panel, percentage of cells with one or more micronuclei from three independent experiments. *$p = 0.011553$ for *lncREST* KD1 vs CTRL HU, *$p = 0.017958$ for *lncREST* KD2 vs CTRL NT, *$p = 0.016264$ for *lncREST* KD2 vs CTRL HU, (mean ± SD, two-tailed unpaired t-test. $n = 3$ per group). **e** Western blot of indicated proteins in CTRL and *lncREST* KD cells with two LNA GapmeRs treated with 1 mM HU o.n. Numbers indicate relative protein quantity normalized to GAPDH. Experiments were performed at least two times for each antibody indicated with similar results. Source data are provided as a Source Data file. **f, g** Representative images and quantification (dotplot and median) of γH2A.X **f** and p-ATR **g** foci per nucleus in HCT116 cells transfected with CTRL or *lncREST* LNAs in NT cells and after treatment with HU 1 mM 3 h and o.n. Scale bar, 10 μm. At least one hundred cells were analyzed per sample in two or three independent replicates. ***$p = 0.0004$ for lncREST KD2 vs CTRL NT γH2A.X, ****$p < 0.0001$ (mean ± SD, two-tailed Mann Whitney U-test). For **a, c–f**, and **h**, source data are provided as a Source Data file.

### *lncREST* regulates replication fork progression

ATR phosphorylation, which is impaired in *lncREST* depleted cells, is one of the first responders to the accumulation of ssDNA stretches in presence of HU induced dNTP depletion[27]. The primary response to RS is the local stalling of forks[14]. To explore the involvement of *lncREST* in this process, we performed fiber assays in *lncREST* knockdown cells. We subjected these cells to two consecutive 20-minute pulses of CldU and IdU, with low doses of HU (50 μM) added during the second pulse to induce the initial local response (Fig. 4a). We found that, while *lncREST* knockdown did not significantly alter the length of DNA fibers under basal conditions, the cells were unable to reduce fork speed in response to HU treatment, indicating an impaired activation of the replication stress surveillance checkpoint (Fig. 4a, b and Supplementary Fig. 4a). We hypothesized that the sustained fork progression in *lncREST* knockdown cells could be linked to discontinuous replication, which would explain the genomic instability observed even in the absence of significant γH2A.X signaling (Fig. 3c–e). To test this hypothesis, we applied a modified fiber assay that involves treating cells with S1 endonuclease before lysis and DNA stretching. This enables the identification of ssDNA gaps along the DNA fiber, as S1 endonuclease specifically cuts them, leading to a shorter fiber tract. Interestingly, in the presence of S1 endonuclease, the pattern of fiber length was reversed in *lncREST* knockdown cells, suggesting that these cells do not respond appropriately to replication stress, and that the DNA fiber is compromised by discontinuous fork progression (Fig. 4c). *LncREST*-depleted cells showed longer inter-origin distance following HU pulses compared to the control cells, in line with an impaired stress signaling (Supplementary Fig. 4b). Additionally, control cells showed a certain level of fork asymmetry, indicating that the replication fork stalling is occurring, but the symmetry is maintained in *lncREST* depleted cells (Supplementary Fig. 3c). These data indicate that *lncREST* is essential for an effective response to replication stress at the fork level and, in its absence, cells bypass the checkpoint and continue replication at damaged forks. Notably, re-expression of *lncREST* in *lncREST* KO cells resulted in a restoration of normal fork progression phenotype, while an empty plasmid had no effect (Fig. 4d, e). These findings highlight the critical role of *lncREST* in ensuring proper replication fork progression and checkpoint activation.

### *lncREST* interacts with replication factors during replication stress

Our findings suggest that *lncREST*, as a chromatin-associated long non-coding RNA with expression peaking in S-phase and further induced upon replication stress, could perform its functions at sites of replication. To explore this possibility, we adapted the iPOND protocol, originally designed to identify proteins associated with replicating DNA, in basal conditions or under replication stress inflicted with HU treatment[28]. The protocol leverages the labeling of short fragments of nascent DNA with EdU, a nucleoside analog of thymidine, and applies click reaction to link biotins for streptavidin pulldown (Fig. 5a). We synchronized cells in S phase (Supplementary Fig. 5a) and treated them with a short pulse of EdU (10 min) to label the nascent DNA. Subsequently, protein-DNA-RNA complexes were crosslinked, bound to biotin with click reaction and followed by streptavidin pulldown (Fig. 5a). To ensure the specificity of the results, we also included a control consisting of a long thymidine chase condition, in which cells were first incubated with EdU for 10 min and then treated with thymidine for 45 min before crosslinking, click reaction and collection (Fig. 5a). In this way, molecules associated with the replication fork are detected only in the EdU short pulse sample, while those obtained after the 45 min thymidine chase are considered chromatin-bound but not specific to the replisome[28]. To ensure that the protocol was working as intended, we performed western blot on the different fractions obtained (Fig. 5b). As expected, the replication factor PCNA was enriched at the forks (EdU pulse without thymidine chase), whereas the histone H3 was mainly detected in the mature chromatin (EdU pulse followed by long thymidine chase) (Fig. 5b). Additionally, when cells were treated with 3 mM HU to induce replication stress and fork stalling, the repair factor RAD51 was observed, as expected, to be enriched at the forks (Fig. 5b). Most interestingly, the analysis of the RNA co-purified with the different fractions revealed a significant enrichment of *lncREST* at the fork compared to the mature chromatin, enrichment that was more pronounced in conditions of replication stress (Fig. 5c). As controls, we used two abundant nuclear lncRNAs, *Malat1* and *Neat1*, known to be enriched in trans at active chromatin sites and well-known components of paraspeckles[29], which both show to be more enriched in the mature chromatin fraction in both non-treated and HU-treated conditions (Fig. 5c and Supplementary Fig. 5b). These observations suggest that *lncREST* preferentially associates with the chromatin at the proximity of stalled forks.

In an attempt to better understand *lncREST* function, we set to identify its protein interactors by pulling down the endogenous lncRNA associated to UV-crosslinked proteins, followed by mass spectrometry. Biotinylated probes were designed to specifically pulldown *lncREST*, while non-targeting lacZ probes were used as negative controls (Supplementary Fig. 5c and Fig. 5d). The experiments did not retrieve *lncREST*-specific interactors in untreated HCT116 cells. On the other hand, when cells were treated with HU, 76 proteins were found specifically associated to *lncREST* in two independent replicates (Supplementary Fig. 5d) (Supplementary Data 2). 70% of these proteins have an exclusive nuclear localization or shuttle from the cytoplasm to the nucleus, and about a third of the nuclear interactors identified are replication factors (e.g. RPA, MCM7, PCNA) or have been identified on nascent DNA in several reports (Fig. 5e)[30–34]. Of note, among *lncREST* nuclear interactors, NCL (Nucleolin) is present with the highest number of peptides (Fig. 5f and Supplementary Fig. 5d).

Together, these results orthogonally demonstrate the association of *lncREST* with factors that participate in the local response to

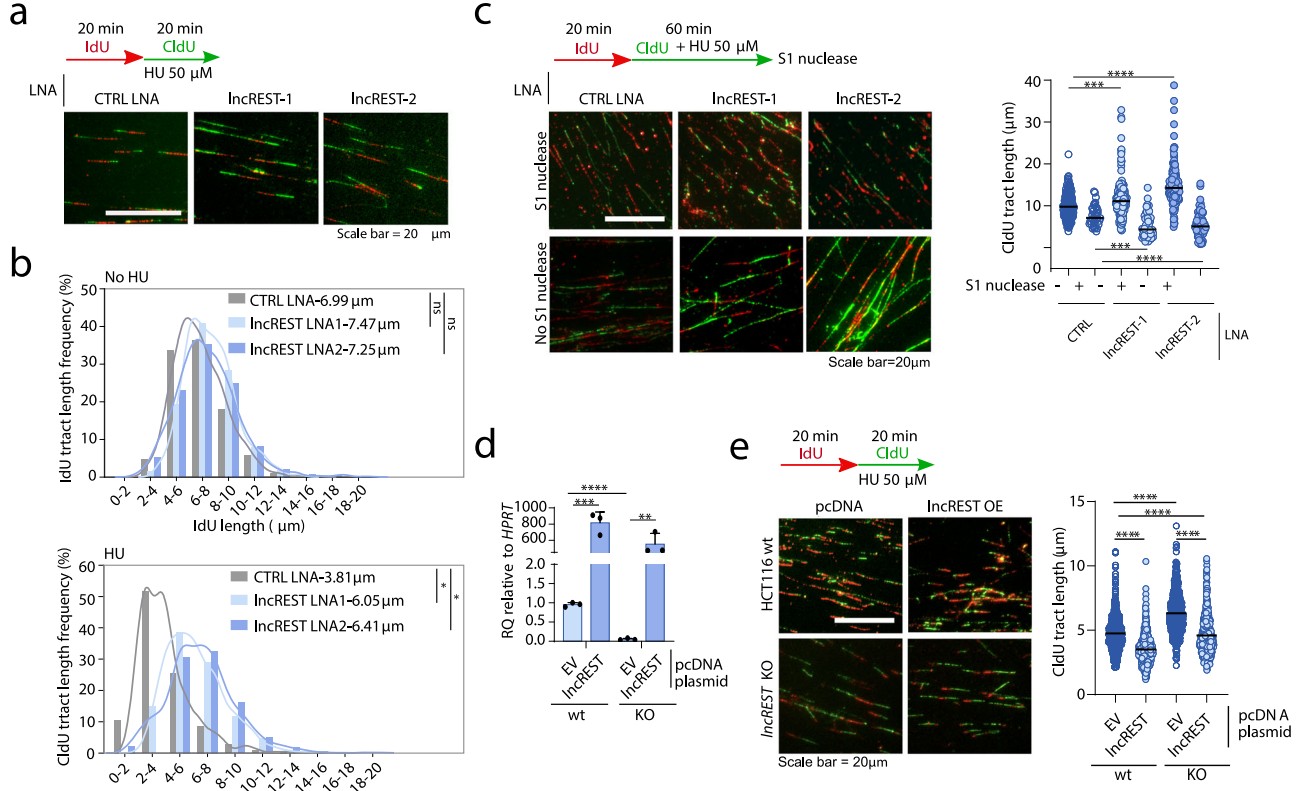

**Fig. 4 | *lncREST* regulates fork progression during replication stress.**
**a** Experimental setup for replication fork analysis in *lncREST* KD cells, and representative fields of DNA fibers immunofluorescence. Scalebar: 20 μm. **b** Tract length frequency analysis in the IdU pulse (left) and CldU + HU (right). For each condition, Kernel density estimates were plotted. At least 250 fibers for each sample were analyzed. *p = 0.0122 in CTRL vs *lncREST* LNA1 CldU; *p = 0.0386 in CTRL vs *lncREST* LNA2 CldU (two-tailed unpaired t-test performed on average tract size for the different replicates; n = 3 per group). **c** Experimental setup for replication fork analysis of control or *lncREST* KD cells treated with S1 nuclease, and dot plot and median of CldU/IdU ratio from 40 to 200 fibers per replicate, n = 3. ***p < 0.001,

****p < 0.0001, by two-tailed Mann-Whitney test. **d** qRT-PCR showing *lncREST* expression in HCT116 wt and lncREST-KO1 cells treated with EV (pcDNA) or pcDNA-lncREST plasmid. **p < 0.01, ***p < 0.001, ****p < 0.0001 (mean ± SD, two-tailed unpaired t-test. n = 3 per group). **e** Experimental setting for replication fork analysis and representative image of fiber assay in HCT116 wt and *lncREST*-KO cells treated with EV (pcDNA) or pcDNA-lncREST plasmid (left), and dot blot with median of IdU (green) tracts (right); at least 150 fibers for each replicate were scored (n = 3) ****p < 0.0001, by two-tailed Mann-Whitney test. Source data are provided as a Source Data file.

replication stress, which could be the molecular mediators of *lncREST* function in this pathway.

## *lncREST* interacts with NCL and is required for RPA deposition on chromatin during replication stress

We identified several replication-related factors interacting with *lncREST*, including the protein NCL. NCL is mainly localized in the nucleolus, but upon replication stress it is recruited at sites of DNA damage, where it participates in the stabilization of stalled replication forks and the activation of DNA repair pathways[35–37]. Therefore, we set to investigate the connection between *lncREST* and NCL. First, we independently confirmed the specific and direct interaction of *lncREST* with NCL by RNA-immunoprecipitation after crosslinking with UV light (UV-RIP) in HU-treated cells (Fig. 6a). We also mapped the region of *lncREST* involved in NCL binding by applying in vitro RNA pulldown assays with different fragments of *lncREST*, including its antisense sequence as control. The assays showed a preferential binding of NCL on the 3' end of *lncREST*, while no interaction was detected with the antisense RNA sequence (Fig. 6b). Together these results further confirmed that *lncREST* and NCL interact in cells undergoing replication stress.

Since the interaction between *lncREST* and NCL was detected when cells were challenged with hydroxyurea (Fig. 5f and 6a), we hypothesized that the lncRNA could regulate NCL function in response to stress. To investigate this functional relationship, we first assessed that the association of NCL with chromatin during RS induced by HU

significantly increases in HCT116 cells, without affecting the total protein levels (Fig. 6c and Supplementary Fig. 6a, b). Subsequently, we analyzed changes in NCL ability to interact with chromatin in *lncREST*-depleted cells, observing a reduced association when KD cells are treated with HU (Fig. 6d). Again, we confirmed that the impaired association of NCL to chromatin is not due to a reduction of the total NCL but rather a mis-localization. More interestingly, the analysis of NCL distribution in the same experimental conditions showed a reduced NCL ratio between the nucleoplasm and the nucleoli (Fig. 6e). We thus concluded that *lncREST* could regulate the availability of NCL at the damaged chromatin following replication stress, therefore affect the function of the protein in this context.

Upon stress and relocation from the nucleolus to other locations in the nucleus, NCL interacts with the replication machinery to regulate it, acting as a protein chaperone[38,39]. Among its known interactors is the RPA complex, which binds to ssDNA and recruits p-ATR to sites of RS, recruitment that is necessary to initiate the replication stress signaling pathway. We have found this specific signaling pathway to be inefficiently activated in *lncREST*-depleted cells (Fig. 3e and g). We therefore investigated whether the reduction of NCL nuclear levels caused by *lncREST* depletion would also be reflected in an impaired RPA deposition. Indeed, cells with decreased levels of *lncREST* presented a reduced recruitment of RPA32 to the chromatin in conditions of HU treatment (Fig. 6d), while the total levels of RPA32 are not affected (Supplementary Fig. 6c). We confirmed this result by performing

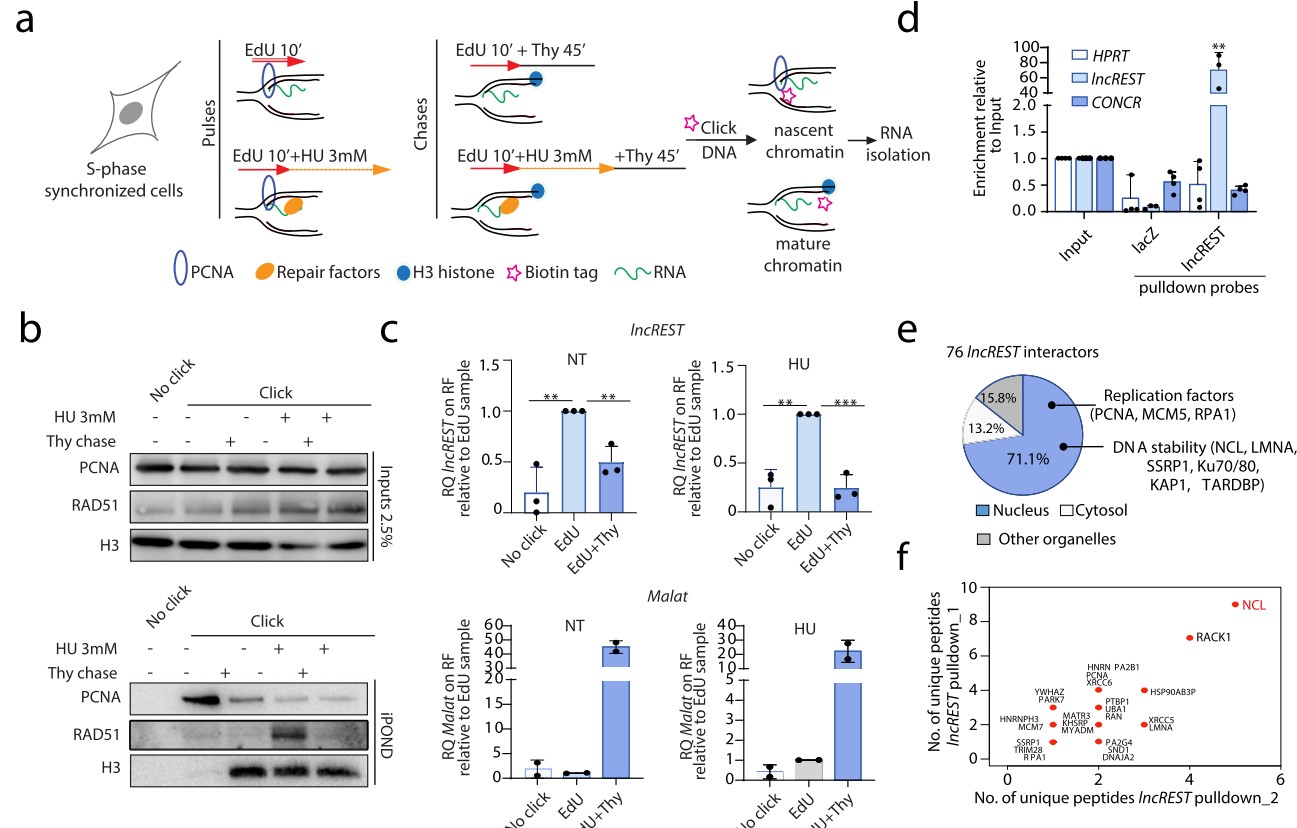

**Fig. 5 | *lncREST* interacts with the replication fork and replication factors.**
**a** Schematic of the adapted iPOND protocol applied to pulldown the RNAs associated to the replication fork. **b** Western blot analysis of the fractions obtained from the different pulses with markers of nascent chromatin (PCNA), mature chromatin (H3) and DNA repair (RAD51). A representative experiment of at least three repetitions. Source data are provided as a Source Data file. **c** qRT-PCR showing the amount of *lncREST* and *MALAT1* relative to the EdU sample (nascent chromatin) in NT (left) and HU treated (right) samples. Each sample was normalized to its input. **p = 0.0052 EdU vs no click NT, **p = 0.0051 EdU+Thy vs EdU, **p = 0.0021 EdU vs

no click HU, ***p = 0.0006 EdU+Thy vs EdU HU (mean ± SD, two-tailed unpaired t-test. n = 3). **d** qRT-PCR showing the efficiency of *lncREST* pulldown by specific biotinylated probes with respect to LacZ probes. The lncRNA *CONCR* and *HPRT* were used as negative controls. **p = 0.0013 *lncREST* enrichment in pulldown; *p < 0.001 (mean ± SD, two-tailed unpaired t-test. n = 3). **e** Pie chart showing the percentage of proteins identified as *lncREST* interactors in the different cellular fractions. **f** Dot plot indicating the number of peptides detected of each of the proteins identified as *lncREST* interactors in two independent experiments. For **b**–**f**, source data are provided as a Source Data file.

immunofluorescence of chromatin-associated RPA32 in *lncREST* depleted cells, after removal of the soluble cellular fraction before fixing the cells (see Materials and Methods) (Fig. 6f). Of note, also RPA1, another subunit of the RPA complex, significantly dropped on the chromatin fraction in *lncREST*-depleted cells treated with HU, indicating that the association of the whole RPA complex might be affected (Supplementary Fig. 6d). RPA reduction in *lncREST* knockdown is likely related to the decreased NCL nuclear localization, as the siRNA-mediated depletion of NCL also resulted in reduced RPA association to chromatin (Supplementary Fig. 6e–g). Accordingly, the knockdown of NCL showed similar signaling defects observed in *lncREST*-deficient cells, since both γH2A.X and p-ATR were reduced compared to control cells despite the increased level of DNA breaks (Supplementary Fig. 6h–j). Therefore, the observed *lncREST*-dependent phenotypes could, at least in part, be associated to deficient localization of NCL at sites of damage outside the nucleolus, required for RPA association to chromatin.

In this context, we hypothesized that *lncREST* could be important for NCL-RPA interactions and subsequent DNA damage signaling. Indeed, this notion was further supported by the decreased co-immunoprecipitation between RPA and NCL from nuclear lysates of *lncREST* knockdown cells in non-treated, but most notably, in HU treated cells (Fig. 6g). We observed similar results in *lncREST* CRISPR KO clones

following HU treatment (Supplementary Fig. 6k), further suggesting that *lncREST* is required for effective interaction between NCL and RPA.

Collectively, these data support a model where *lncREST* favors the localization of factors at the stressed forks to coordinately execute an effective stress signaling.

## *lncREST* inhibition causes mitotic defects, increased apoptosis, and impairs tumor growth

Given the role that *lncREST* has in the stress response, we investigated its possible link with cancer, reasoning that its expression could be related to the oncogenic capacity of cancer cells. We therefore evaluated the association of *lncREST* with replication stress phenotypes in tumors from colon cancer patients. To do this, we measured the scores of Chromosomal Instability (CIN) signatures of colon cancer tumors from The Cancer Genome Atlas (TCGA), recently defined based on different types of known driver genes associations and molecular features[40]. After establishing two major patients' cohorts based on *lncREST* expression levels (low or high), we assessed the link between *lncREST* expression and each signature score. Interestingly, out of all the CIN signatures defined in the tumors[40], patients with low *lncREST* expression showed a significantly higher score for CX9, defined as a replication stress signature associated with increased cell cycle score and chromotripsis, a phenomenon strongly related to

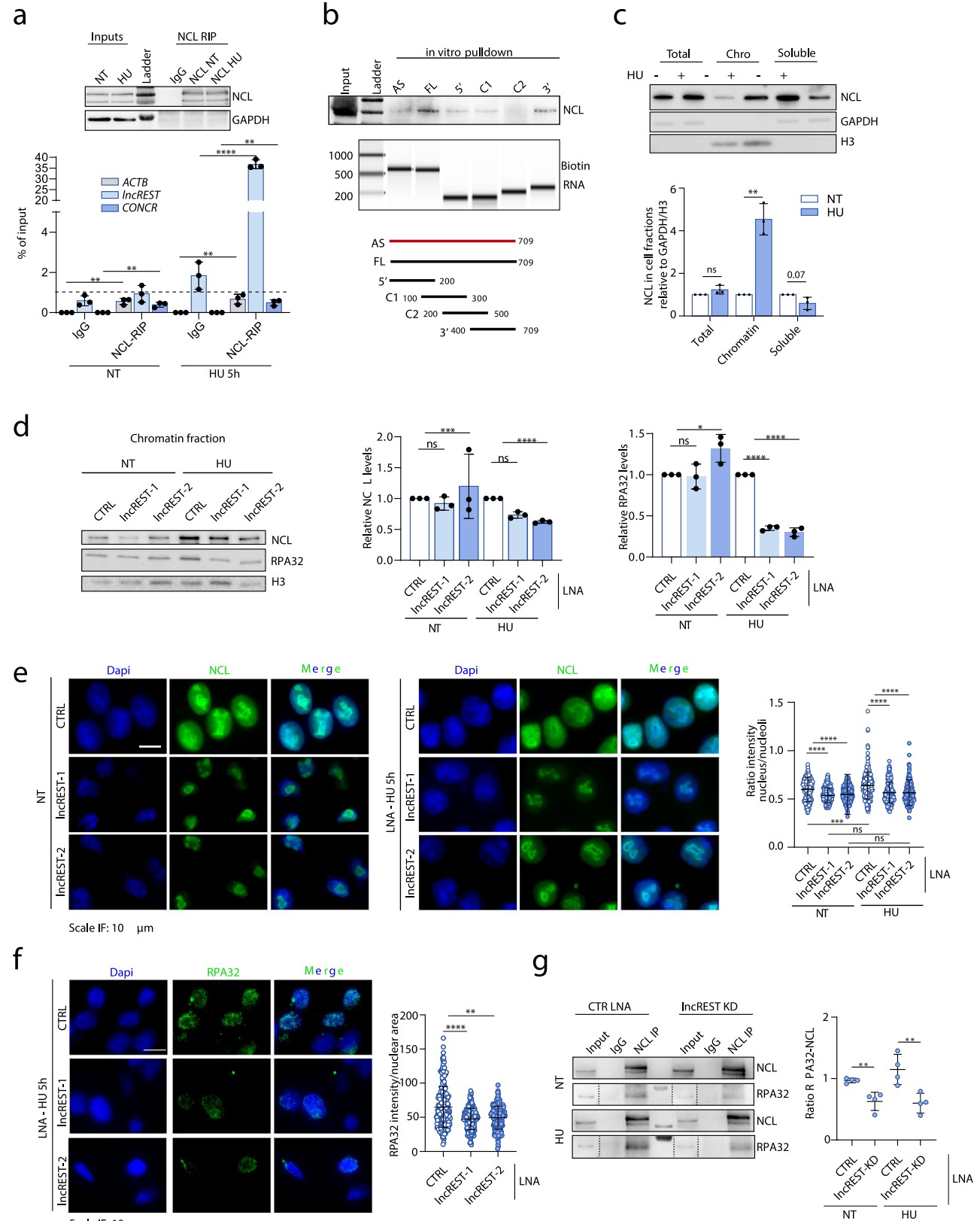

micronuclei formation and defects in chromosome segregation[41] (Fig. 7a, Supplementary Fig. 7a).

We further sought to determine if the defects in the replication stress response arising when *lncREST* levels are reduced could be linked to chromosomal instability phenotypes. We predicted the presence of mitotic defects in *lncREST*-depleted cells, since altered

functions of ATR and NCL have been directly associated to this type of alterations[42,43]. As expected, we identified a significant impairment of chromosome congression, observing two major alterations: misalignment (1 to 10 chromosomes failing to align on the mitotic spindle), and non-alignment (more than 10 dispersed chromosomes) (Fig. 7b). This phenotype, together with the defective activation of replication stress

**Fig. 6 | *lncREST* interacts with NCL and modulates RPA association to chromatin. a** RNA immunoprecipitation with NCL antibody shows *lncREST*-NCL association. The upper panel shows successful NCL pulldown by western blot in NT and HU-treated cellular lysates, and relative inputs. The lower panel shows qRT-PCR detection of *lncREST* precipitated by NCL. **$p = 0.0093$ *ACTB* HU, **$p = 0.0047$ *CONCR* HU, ****$p < 0.0001$ *lncREST* HU (mean ± SD, two-tailed unpaired t-test. $n = 3$). **b** In vitro RNA pulldown assay revealed a preferential binding of NCL to the *lncREST* fragment containing the 3′ end. Antisense *lncREST* sequence was used as control. Results were confirmed by three independent experiments. **c** Upper panel, Western blot of NCL in cellular fractions of HCT116 NT and treated with 1 mM HU o.n. Bottom panel, quantification of NCL on each fraction relative to H3 or GAPDH. **$p = 0.007081$ (mean ± SD, two-tailed unpaired t-test. $n = 3$). **d** Left panel, Western blot of chromatin fractions of *lncREST* depleted HCT116 cells shows reduction of chromatin-associated NCL and RPA32 in 1 mM HU treated cells. Right panel, quantification of NCL and RPA32 on chromatin fraction relative to H3. For NCL

**$p = 0.001745$, ****$p < 0.0001$; for RPA32 *$p = 0.030133$, ****$p < 0.0001$ (mean ± SD, two-tailed unpaired t-test. $n = 3$). **e** IF of NCL in *lncREST* depleted and control HCT116 cells in NT and HU treated condition (2 mM for 5 h) showing a reduction of the nuclear pool of NCL in KD cells. ***$p = 0.0005$, ****$p < 0.0001$ (mean ± SD, two-tailed unpaired t-test. $n = 3$). **f** Immunofluorescence of chromatin-associated RPA32 of HCT116 transfected with CTRL and *lncREST* LNA GapmeRs, following HU treatment as indicated, after removal of soluble cellular fraction before fixing the cells, and dot blot (two-tailed unpaired t-test, mean ± SD) of RPA32 fluorescence normalized by nuclear area. At least 70 cells for each biological replicate were analyzed. **$p = 0.0086$, ****$p < 0.0001$ (two-tailed unpaired t-test. $n = 3$ per group). **g** Western blot (left) and quantification (right) of CTRL LNA or *lncREST* KD cells- NCL co-immunoprecipitated with RPA32 or a control IgG in untreated or HU-treated cells. **$p = 0.0043$ in NT, **$p = 0.0098$ in HU (mean ± SD, two-tailed unpaired t-test. $n = 4$). Source data are provided as a Source Data file.

signaling and accumulation of DNA damage, was mirrored by NCL KD (Fig. 6, Supplementary Figs. 6 and 7b)[43]. Consequently, the inhibition of *lncREST* also resulted in decreased proliferation of colorectal cancer cells, more pronounced when further challenged with hydroxyurea (Fig. 7c). Similar delay in cell proliferation related to a deficient expression of *lncREST* was observed in cells where *lncREST* gene had been genetically removed (Fig. 7d). Moreover, the downregulation or knockout of *lncREST* reduced the capacity of the cells to form tumors when orthotopically injected in mice (Fig. 7e, f and Supplementary Fig. 7c), which supports that these cells require *lncREST* expression to proliferate in vivo. When analyzing in more detail the cell proliferation defects, we observed that *lncREST*-depleted cells presented increased levels of apoptosis (Supplementary Fig. 7d), likely related to the genomic instability, and impaired stress signaling caused by *lncREST* depletion (Fig. 3).

Together, these results suggest that *lncREST* deficiency-induced genomic instability, mitotic errors, and consequently, decreased tumorgenicity of cancer cells, likely result from a compromised NCL availability and lead to a defective response to replication stress.

## Discussion

The response to replication stress involves a complex interplay between various cellular processes, including DNA repair, checkpoint activation, and chromatin remodeling. Its main players must be regulated in a quick and coordinated fashion to provide an effective protection of the cell's genomic integrity. Here we show that, besides the well-known protein elements of the replication stress pathway, this response has a transcriptional component that involves the induction of chromatin-associated lncRNAs controlled by stress transcription factors. Among them, we uncover the function of the previously uncharacterized long non-coding RNA *lncREST*. We show that *lncREST* is necessary for a correct establishment of the replication stress response, and we molecularly characterize the impaired activation of the S-phase checkpoint signaling in *lncREST* depleted cells when challenged with hydroxyurea.

*LncREST* is transcriptionally controlled by p53, assuring its fast activation in response to DNA damage. It is also located in a head-to-head configuration with the protein-coding gene *TP53INP1*.

Both genes are co-regulated at the transcriptional level, sharing regulatory elements. Our data also suggest that *lncREST* might influence *TP53INP1* expression, varying in opposite directions depending on the specific loss-of-function method employed to perturb *lncREST* expression. While the exact mechanism underlying this regulation requires further investigation, it's noteworthy that the role of *lncREST* in the replication stress response appears to be independent of the *TP53INP1* gene.

This conclusion is supported by several evidence: (i) the observed enrichment of *lncREST* at replication forks, (ii) the lncRNA's ability to rescue abnormal fork progression when expressed in trans, and (iii) its

dependence on the interaction with the protein NCL, which is part of the same pathway. Nevertheless, it is possible that *lncREST* could have additional biological effects through the regulation of *TP53INP1* expression. *TP53INP1* encodes a component of the autophagy pathway with distinct cellular activities[25], and the co-regulation of *lncREST* and *TP53INP1* hints at potential connections between the replication stress and autophagy responses, which remain to be uncovered.

Here, by using a set of orthogonal experiments, we unfolded the role of *lncREST* in the replication stress response. One of the key features of this lncRNA is its localization on the chromatin, which we identified biochemically and visualized by RNA-FISH. Most interestingly, the analysis of the RNA co-localizing with nascent DNA and the unbiased identification of the proteins co-purifying with *lncREST*, revealed that it is enriched at stressed forks together with a set of proteins that are known players in this mechanism.

The most abundant *lncREST* interactor identified was nucleolin (NCL), an RNA-binding protein part of the fibrillar layer of the nucleolus[44,45]. However, the nucleolus is not NCL's exclusive localization. NCL, as a multidomain protein with chaperon-like activity, has been implicated in a wide range of cellular processes, including ribosome biogenesis, gene expression regulation, chromatin remodeling, and replication stress. Although the exact mechanisms by which NCL contributes to the replication stress response remain to be fully understood, NCL changes its localization upon stress and interacts with several factors[35–37]. Of note, in mouse stem cells NCL had been reported to interact with the nucleolar lncRNA *Discn*, increasing NCL nucleolar retention and regulating its role in the stress response[12]. Remarkably, our FISH experiments show that *lncREST* forms discrete foci evenly distributed in the nucleus, without any specific nucleolar-like distribution. Its depletion impairs NCL re-localization, reducing its interaction with RPA and resulting in defective replication stress signaling. NCL, which can establish multiple interactions with RNAs and proteins, contains two intrinsically disordered regions (IDRs) that confer it with the capacity to form biomolecular condensates[46]. We speculate that this property may be key for NCL function in replication stress, particularly in relation to *lncREST*. This is based on the notion that chromatin-associated lncRNAs are known to nucleate factors in the form of condensates[4,47], and growing evidence supports that biomolecular condensates are functionally formed at the sites of damage[5,48]. *LncREST* may be one of the components of such condensates, favoring their formation involving interactions with NCL and possibly other proteins. This is further corroborated by the ability of *lncREST* to co-purify together with components of the replication stress response other than NCL, which are known to interact with each other in the establishment of the replication stress response. It should also be noted that *lncREST* sequence contains a SINE repeat, a type of sequence strongly enriched in RNAs within phase-separated coacervate microdroplets[6]. We envision that besides *lncREST*, additional ncRNAs are part of such subnuclear phases, helping orchestrate the

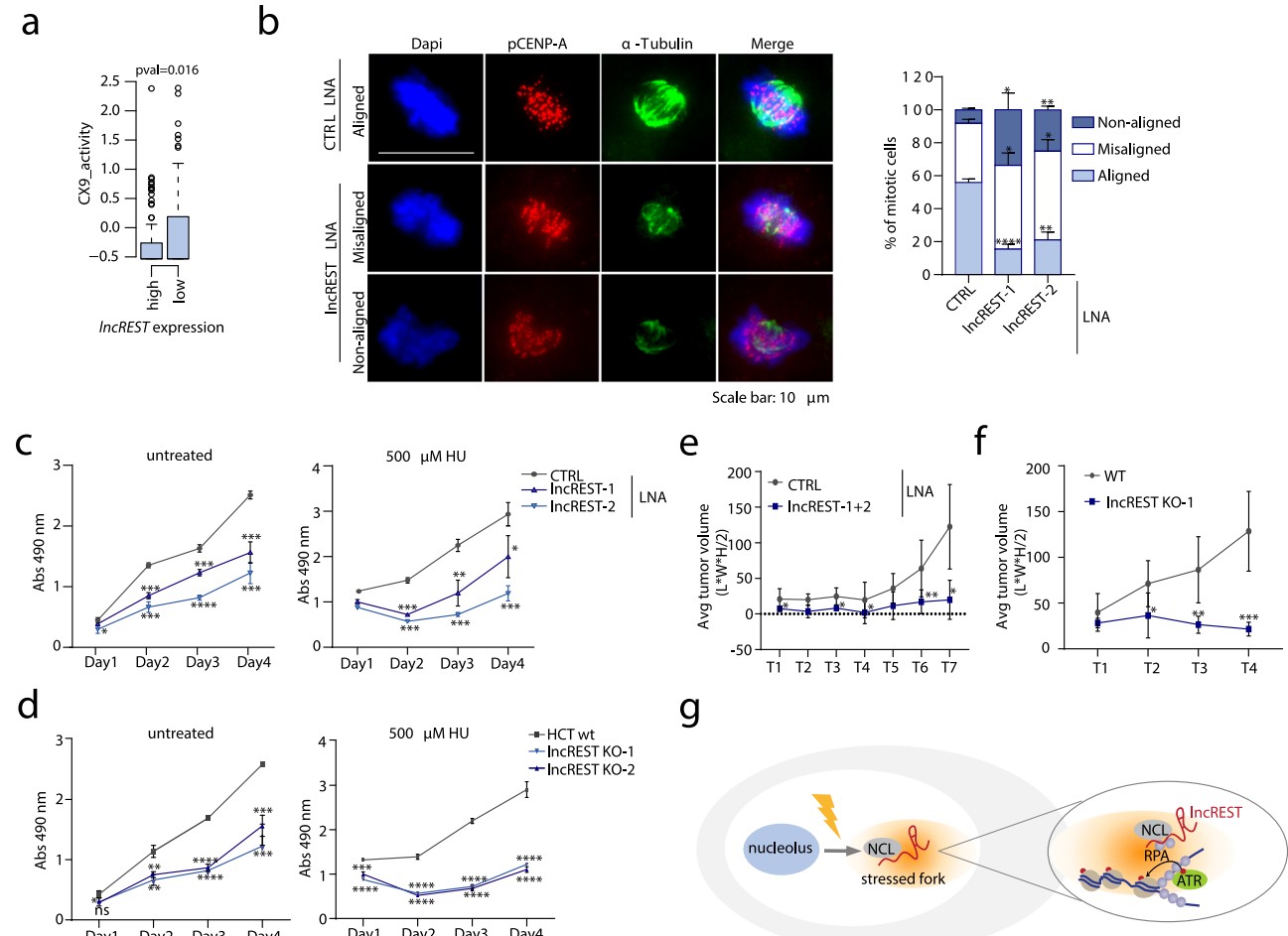

**Fig. 7 | *lncREST* regulates proliferative phenotypes and is required for accurate mitosis. a** Score of the replication stress CX9 signature[40] in TCGA colorectal tumors with high and low *lncREST* expression. TCGA patient samples were divided in two groups depending on their *lncREST* expression (40% upper percentile as high, 60% lower percentile as low). For each one of them, the activity of the 17 copy number signatures was calculated and plotted. Boxplots represent 25 to 75 percentiles, whiskers are 1.5 x interquartile range (interquartile range = percentile75-percentile25). **b** Mitotic HCT116 cells stained with α-tubulin (green) and p-CENP-A (red) antibodies and transfected with CTRL or *lncREST* targeting LNA GapmeRs, and relative quantification showing the percentage of mitotic cells identified for each type of alignment. Representative image of the predominant type of alignment for CTRL and *lncREST* KD cells, are shown. *n* = 3 for CTRL and *lncREST* KD1, 2 for *lncREST* KD2. *p < 0.05; **p < 0.01; ***p < 0.001. (mean with SD, two-tailed unpaired t-test). **c, d** Cell proliferation measured my MTS assay in HCT116 cells (untreated and HU-treated) transfected with *lncREST* of CTRL LNA

GapmeRs **c** and *lncREST* KO (two-tailed unpaired t-test, mean ± SD, *n* = 3). *p < 0.05; **p < 0.01; ***p < 0.001, ****p < 0.0001 **d**. Absorbance (Abs) at 490 nm was measured over a time course of 4 days, starting from 48 h post transfection in (**c**), and the day after plating the cells in (**d**). *p < 0.05; **p < 0.01; ***p < 0.001; (mean ± SD, *n* = 3 per group). **e, f** Analysis of tumors generated by subcutaneous injection of control or *lncREST*-depleted HCT116 cells **e** or *lncREST* KO cells **f** in BALB/cA-Rag2 − /−γc − /− mice. Graphs show average tumor volume ±STD calculated using the formula Length x Width x Height/2. two-tailed unpaired t-test (*n* = 4 for **e**, *p = 0.0012 T1, *p = 0.035 T3, *p = 0.027 T4, **p = 0.017 T6, *p = 0.049 T7. *n* = 6 for **f**, *p = 0.036 T2, **p = 0.0028 T3, ***p = 0.00015 T4. **g** Proposed model for the mechanism of action of lncREST. Upon replication stress, NCL protein localizes to the chromatin where it interacts with lncREST. This is required for the localization of RPA and p-ATR at the forks for the effective S-checkpoint signaling, leading to fork stalling. The action of *lncREST* protects cells from genomic instability by assuring the correct functioning of the S-phase checkpoint. For **b−f**, source data are provided as a Source Data file.

replication stress response. Thus, the discovery of the role of *lncREST* in regulating replication stress suggests that some nuclear RNAs may have a more significant role in maintaining genomic stability than previously anticipated.

## Methods

### Cell culture, interference, plasmids and lentivirus transfections

HCT116 (CCL-247), H358 (CRL-5807), HEK-293T (CRL-3216), LoVo (CCL-229), RKO (CRL-2577), A549 (CRM-CCL−185), RPE1 and U2OS (HTB-96) were purchased from the American Type Culture Collection (ATCC). COV362, purchased from MERK, were from European Collection of Authenticated Cell Culture (ECACC). JHH were obtained from Puri Fortes lab at CIMA (Centre for Applied Medical Research), TIG-3 cells were from Anders Lund's lab at BRIC, University of Copenhagen. HCT116, H358, LoVo, RKO and JHH6 cells were

maintained in RPMI 1640 (GIBCO), HEK-293T, A549, RPE1, TIG3 and COV362 cells were maintained DMEM (GIBCO) supplemented with 10% FBS (GIBCO) and 1% penicillin/streptomycin (Lonza) in 5% $CO_2$, at 37 °C. LNA GapmeRs against *lncREST* and LNA control were designed and purchased from QIAGEN, and were used at a final concentration of 4 nM, siRNAs against *NCL* and control siRNA were designed with iScore designer tool[49] and purchased from Sigma and used at a final concentration of 40 nM. All RNA interference experiments were performed using Lipofectamine RNAiMAX (Life Technologies) according to the manufacturer's instructions. pcDNA3.1(+) vector containing full-length *lncREST* was purchased from GenScript. For plasmid transfections and lentivirus production in HEK-293T, Lipofectamine 2000 (Invitrogen) was used according to the manufacturer's protocol. To generate stable cells for DCas9/KRAB studies, HCT116 wt or dCAS9-P2A-RFP cells were infected with described

lentiviruses (see list in Supplementary Data 3) by using 4 µg/ml polybrene (Santa Cruz).

### CRISPR/Cas9-mediated interference and knockouts

The single-guide RNAs (sgRNA) used for dCas9/KRAB or knockout of *lncREST* were designed using the Benchling tool (www.benchling. com). For CRISPR KO, two sgRNAs targeting the entire sequence of *lncREST* were designed. The 3' sgRNA was cloned into a CRISPseq-BFP plasmid (Addgene #85707). HEK-293T were transfected with PMD2.6, psPAX2, and sgRNA-CRISPseq-BFP to generate lentiviral particles that were used to generate stable 3'sgRNA-KO9-BFP HCT116. BFP-positive cells were sorted 5 days post transfection. The 5'sgRNA was cloned into a pX459 plasmid (Addgene, #48139)[50], transfected into 3'sgRNA-KO9-BFP-HCT116 cells. At 48 h post-transfection, cells were selected with puromycin. After selection, single cells were sorted into 96-well plates. Single colonies were grown and screened for loss of *lncREST* by PCR. For DCas9/KRAB, HEK-293T were transfected with pMD2G (Addgene, #12259), psPAX2 (Addgene, #12260) and pHR-SFFV-KRAB-dCas9-P2A-mCherry (Addgene, #60954)[23] plasmids to generate lentiviral particles, that were used to transfect HCT116 cells. 5 days post-transfection, RFP-positive cells were sorted to obtain a stable cell line. Two independent sgRNAs designed in the TSS of *lncREST* were cloned into a KO9-BFP plasmid, and lentiviral particles generated into HEK-293T cells were infected into the stable dCAS9-P2A-RFP HCT116 cells. Five days post infection, the pools of BFP-positive cells were selected by sorting, and *lncREST* depletion was tested by RT-qPCR.

### Drug treatments and cell culture supplements

HU (Sigma-Aldrich, H8627-5G), CPT (Sigma-Aldrich, C9911–100MG), cisplatin (Merck KGaA, Darmstadt, Germany,60778-25EA), doxorubicin (Sigma-Aldrich, D1515–10MG), nutlin (Sigma-Aldrich, N6287–1MG) were applied as indicated. dNTPs analogs CldU (Sigma-Aldrich, C6891–100MG), IdU (Sigma-Aldrich, I7125-5G), and thymidine (Sigma-Aldrich,89270-5 G) were applied as indicated.

### Cell fractionation

A total of $1 \times 10^7$ HCT116 cells were harvested with trypsin after 8 hrs of treatment with 1 mM of HU, or PBS and washed with ice-cold PBS. One-tenth aliquot of cells was put aside for total RNA isolation, the remaining was used for cell fractionation, according to ref. 15. All steps were performed in the presence of protease inhibitors (Roche), phosphatase inhibitors (Roche), and RNAsin (Promega). Cells were resuspended in 200 µl of isotonic lysis buffer (10 mM Tris-HCl pH7, 150 mM NaCl, 0.15% NP-40) for 5 min on ice and layered on a sucrose buffer (10 mM Tris-HCl, 150 mM NaCl, 25% sucrose). Nuclei were centrifuged for 10 min at full speed to recover the supernatant as the cytoplasmic fraction. The nuclear pellet was washed (1 mM EDTA, 0.1% Triton-X100 in PBS), and resuspended in 200 µl glycerol buffer (20 mM Tris-HCl pH8, 75 mM NaCl, 0.5 mM EDTA, 505 glycerol, 0.85 mM DTT) and finally lysed with 200 µl of nuclear lysis buffer (20 mM HEPES, 300 mM NaCl, 1 M urea, 0,2 mM EDTA, 1% NP-40, 1 mM DTT). Lysed nuclei were centrifuged at full speed for 2 min to separate the soluble fraction (supernatant) from the chromatin-associated fraction (pellet).

### RNA isolation and qRT-PCR

Maxwell RSC Simply RNA kit (Promega) was used to isolate RNA from total, cytoplasmic, nuclear soluble and chromatin fractions used for RNA-seq and relative validations. Total RNA from cultured cell lines with different treatments was extracted with TRIzol Reagent (Invitrogen) according to the manufacturer's protocol. For qRT-PCR, cDNA synthesis was performed using the high-capacity RNA to cDNA kit (Invitrogen). qPCR was performed using SYBR green master mix (BioRad) in a Quantstudio 3 real-time PCR system (Applied Biosystems). The relative expression of different sets of genes was quantified to HPRT mRNA. QuantStudio Real-Time PCR software v1.6.1 was used

to collect quantitative PCR data. Primer sequences for qRT-PCR are listed in Supplementary Data 3.

### Western blot

Cell lysates from total and fractionated cells were quantified by using the Pierce BCA protein assay kit (Thermo Scientific). Equal amounts of protein were resolved by 10 or 12% Tris-Glycine gels by SDS page. Proteins were blotted into nitrocellulose membrane, blocked with blocking buffer (PBS-Tween 1%, 3% BSA) an incubated with the corresponding primary antibodies (1:1000), and HRP-conjugated secondary antibodies (1:5000). Blots were developed with Western lightning plus-ECL (Perkin Elmer). Images were acquired in an Odyssey XF imaging system (Li-COR) and analyzed with the software Image Studio Lite 5.2. All antibodies are listed in Supplementary Data 3.

### Library preparation, RNA-seq and data analysis

For RNA-seq of total and chromatin fractions of HCT116 cells non-treated and treated with 1 mM HU for 8 h, RNA was isolated and DNAse I-treated with Maxwell RSC Simply RNA kit (Promega). After quality and integrity check with RNA screen tape (Agilent Technologies), RNA was processed for library preparation with TruSeq Stranded Total RNA kit (Illumina), according to the manufacturer's instructions, and sequenced on Illumina Nextseq 500 (40 million reads/sample).

RNA sequences were trimmed with Trimmomatic v0.38. Sequencing data for cellular fraction were aligned to the genome assembly hg38 using STAR[51] with default parameters, assigned to genes with FeatureCounts v1.6.3[52]. Differential expression analysis between HU and non-treated cells was carried out by using edgeR[53] in R/Bioconductor: genes were tested for differential expression only if expression (cpm) was greater than 2 in at least two samples of comparison groups. Significant genes were selected applying the following filters: FDR < 0.05; $|\log_2 FC| > \pm 1$. For RNA-seq analysis of synchronized HCT116 and released in the cell cycle fractions, Fastq files were aligned to the hg19 human with STAR, reads aligning to GL contigs were removed and FeatureCounts was used to quantify the number of reads falling in annotated genes. DESeq2[54] was used to measure different expression genes.

### Cell synchronization

To test the distribution of RNAs in the different cell fractions and for identification of RNA on nascent chromatin, HCT116 were synchronized by double thymidine block as follows. Different plates were seeded at 60% confluency. The next day, medium was replaced with growth medium containing 2 mM of Thymidine and incubated for 16 h. Then, the medium was removed, cells washed with PBS and fresh medium was added. After 9 h, 2 mM thymidine was added again to the medium and incubated for an additional 14 h. At this point, cells synchronized in G1/S were washed with PBS, and normal medium was added to different plates over a time course, cells were collected at different time points and tested by propidium iodide staining to verify their cell cycle position.

### Cell cycle

After cell synchronization/release, cultured cells were trypsinized and washed with PBS. A total of $1 \times 10^6$ cells were centrifuged at 300 x *g* for 5 min and resuspended in 200 µl Ethanol 70% overnight at 4°C. Fixed cells were washed with PBS and resuspended in 250 µl PBS adding 5 µl of 10 mg/ml RNAse A solution for 1 h at 37 °C. After adding propidium iodide cells were analyzed in a CytoFLEX-LX cytometer (Beckman Coulter).

### RNA-FISH

FAM-labeled Locked Nucleic Acid (LNA) DNA probes were synthesized by IDT (sequences provided in Supplementary Data 3.). Unsynchronized HCT116 were cultured on coverslips and following the indicated

treatments they were fixed with 4% paraformaldehyde, washed with PBS and permeabilized with 70% Ethanol overnight. After two 10 min PBS washes, cells were incubated with acetylation buffer (0.1 M Triethanol amine, 0.5% acetic anhydride in $H_2O$) to eliminate RNases. After PBS washes, cells were pre-incubated with hybridization buffer (50% deionized formamide, 2X SSC, 10% dextran sulfate in $H_2O$) at 55 °C for 1 h. Probes were denatured at 92 °C for 4 min and added to the cells overnight at 37 °C. Slides were washed extensive consecutive washes with 2X SSC and 50% formamide at 55 °C (30 min), 2X SSC at 55 °C (30 min), 1X SSC at 37 °C (30 min), PBS (2 washes at RT for 5 min). Cells were incubated in blocking buffer (10% goat serum, 0.5% blocking reagent (Roche) in PBST-0.5% Tween) for 1 h at RT, then anti FAM-POD (Roche) 1:150 diluted in $H_2O$ was added for 1 h. Following 3 washes in 4X SSC in dark for 5 min, cells were incubated with TSA-Cy3 1:600 in amplification diluent (Perkin Elmer), washed 3 times in 4X SSC for 10 min, washed 3 times in 4X SSC-0.1% Triton X−100 for 10 min, washed once with 4X SSC for 10 min, mounted with ProLong glass antifade mountant with nuclear blue (Invitrogen) and imaged in a confocal microscope (Axio Observer. Z1/7, Carl Zeiss) at 63X magnification.

### Immunofluorescence

For immunofluorescence staining, cells were seeded on coverslips and fixed in 4% paraformaldehyde for 15 min at RT, permeabilized in 0.25% Triton X−100 in PBS for 20 min, washed with PBS and blocked with 1% BSA in PBS for 30 min at RT. Primary antibodies were diluted in blocking buffer and applied to the cells for 1 h at RT: anti phospho-ATR (1:350), anti phospho-Histone H2A.X (ser139) 1:500, anti-NCL (1:500), followed by three 5 min washes in PBS. Alexa-fluor conjugated secondary antibodies (Invitrogen) were diluted 1:500 in blocking buffer and applied for 1 hr at room temperature, followed by three 5 min washes in PBS. Cells on coverslips were mounted with ProLong glass antifade mountant with nuclear blue (Invitrogen). For immunofluorescence of DNA fibers, rat monoclonal anti-BRdU (anti-CldU) and mouse monoclonal anti-BrdU (anti-IdU) were used as primary antibodies at 1:100 overnight at 4 °C. The secondary antibodies were AlexaFluor conjugated antibodies (Invitrogen) used at 1:300 for 1 hr at RT. ssDNA primary antibody was used to detect fibers integrity at 1:300 for 30 min, with an AlexaFluor conjugated antibody (1:300). For PCNA immunofluorescence, cells grown on a coverslip were washed with PBS and incubated with hypotonic buffer (10 mM Tris HCl pH 7.4, 2.5 mM MgCl2, 0.5% NP-40) for 10 min at 4 °C. After two washes in cold PBS, cells were fixed in 1% formaldehyde for 5 min with gentle rocking, washed with PBS and post-fixed with ethanol 70% overnight. The next day, cells were washed with PBS, incubated for 15 min in blocking buffer (0.2% Tween, 1% BSA in PBS). Slides were incubated with anti-PCNA antibody 1:100 in blocking buffer for 1 h at room temperature. Following three PBS washes slides were incubated with AlexaFLuor secondary antibody 1:100 in blocking buffer for 1 h. After three final PBS washes, slides were mounted with ProLong glass antifade mountant with nuclear blue (Invitrogen). All samples were imaged in a Zeiss Axio Imager M1 at 63X magnification for protein immunofluorescence, or 20X magnification for the fiber assays.

For RPA32 immunofluorescence on chromatin, cells were incubated with hypotonic buffer (10 mM Tris-HCl pH 7.4, 2.5 mM MgCl2, 0.5% NP40) for 10 min at 4 °C, followed by two PBS washes. Then, cells were fixed with formaldehyde 1% for 5 min at RT, and post-fixed with ethanol 70% overnight. Following PBS washes, slides were blocked with PBST−1% BSA for 15 min, and incubated with primary antibody against RPA32 at 1:100 in blocking buffer for 1 h at RT. Slides were washed for three times (10 min each) with PBS with 0.2% Tween, and incubated with fluorescent secondary antibody (Invitrogen) 1:500 in blocking buffer for 1 h. Slides were mounted as described above and imaged with a 63X objective. Immunofluorescence experiments were analyzed by using ImageJ Fiji

package. For NCL quantification in nucleolus and nucleoplasm, we implemented a macro for Fiji to detect the fluorescence intensity of the nucleoli perimeter, area, and the nuclear area around nucleoli. For mitotic cells analysis, we analyzed chromosomes alignment in prometaphase cells with a clear bipolar spindle. Statistical analysis was performed by using Student's t-test, GraphPad Prism 8.0.2 was used to analyze the statistical significance. *$p < 0.05$; **$p < 0.01$; ***$p < 0.001$; ns, no significant difference.

### Alkaline comet assay

HCT116 cells were transfected with control or *lncREST* targeting LNAs, and *lncREST* KO cells were treated with 1 mM HU overnight or 40 μM APH to induce DSB. Cells were trypsinized, then cells were resuspended in 0.8% low melting agarose (Conda) and layered on cold slides pre-coated with 1% agarose. Cells were lysed overnight at 4 °C in prechilled lysis buffer (2.5 M NaCl, 10 mM Tris-HCl pH7, 100 mM EDTA pH8,1% Triton X−100, 1% DMSO pH 10), rinsed with $H_2O$ and incubated in alkaline buffer (300 mM NaOH pH>13, 1 mM EDTA) for 20 min prior to electrophoresis. Electrophoresis was performed in an alkaline buffer at 1 V/cm with a constant 300 mA current for 30 min at 4 °C. Following electrophoresis, slides were neutralized in Tris-HCl 400 mM pH 7.5 and washed once with $H_2O$. Slides were fixed in 100% ethanol and stained with ProLong glass antifade mountant with nuclear blue (Invitrogen). Comet images were acquired on a Zeiss Axio Imager M1 at 10X magnification. Comet tail moment was determined using OpenComet v1.3.1 (www.cometbio.org) as an ImageJ plugin.

### DNA fiber assay

HCT116 transfected with control or *lncREST* targeting LNAs, and lncREST KO cells transfected with control plasmid or *lncREST* expression plasmid were labeled with 50 μM 5-chloro-2'-deoxyuridine (CldU; Sigma, C6891) for 20 min and then incubated with 150 μM of 5-iodo-2'-deoxyuridine (IdU; Sigma, I7125) for 20 min with 50 μM of hydroxyurea (HU, Sigma). For S1 nuclease experiment, the IdU pulse with HU was 60 min. Cells were then permeabilized with CSK100 buffer and treated with S1 nuclease (20U/ml) for 30 min at 37 °C. Cells were collected and resuspended in cold PBS at $2.5 \times 10^5$ cells/ml and 500 cells were lysed in 10 ul of pre-warmed spreading buffer (0.5%SDS, 200 mM Tris-HCl pH 7.4, 50 mM EDTA) on one end of the glass slide. Slides were tilted to allow the spread of DNA fibers along the glass, fixed with freshly made ice-cold methanol:acetic acid 3:1 and treated with 2.5 M HCl. Labeled fibers were detected by immunofluorescence as described above. Immunostained slides were mounted with Prolong glass antifade mountant (Invitrogen) and images were captured with a Zeiss Axio Imager M1at 20X magnification. Fiber length was measured using the ImageJ software, values were converted to kb (1 μm=2.59 kb). At least 200 fibers per sample were analyzed per replicate in the fiber speed analysis. In all experiments, we only measured IdU tracts that were consecutive to a CldU tract.

### Identification of RNA on nascent chromatin

We set out to identify the RNA associated to the replicating DNA in HCT116 non-treated and HU-treated cells, with relative controls (no click negative control, thymidine chase to mark mature chromatin) for a total of 5 conditions. To isolate RNA associated with the replication fork, seven 15 cm plates at 70% confluence per condition were first synchronized with double thymidine block as described above and released in normal medium for three hours, to obtain a population of cells in S-phase. Isolation of the replication fork was performed as already described[28] with some modifications. Briefly, cells were pulsed for 10 min with 10 μM EdU (Invitrogen) to mark nascent DNA. For the HU-treated condition, the EdU treatment was followed by a subsequent 5 h incubation in a medium containing 3 mM HU and 10 μM EdU. For the chase conditions (for both non-treated and treated samples), the EdU treatment was followed by two quick PBS

washes and incubation with a medium containing 10 μM of Thymidine (Sigma) for 45 min. Following the chases, cells were washed with PBS and crosslinked with 1% formaldehyde for 15 min. Cells were scraped and permeabilized with 0.25% Triton X−100 in PBS. Click reaction was performed on the washed permeabilized cells, adding RNAsin (Promega) to the click-it reaction buffer (1 μl/ml). Click reaction was performed at RT for 1 h under rotation. From this point on all the steps were performed in presence of protease inhibitors, phosphatase inhibitors and RNAse inhibitors and at 4 °C. Cells were lysed (1%SDS, 50 mM Tris-HCl pH 8) and sonicated. Inputs were taken at this point, two inputs for each sample for protein and RNA, respectively. EdU-biotin labeled DNA/protein/RNA was pulled down with streptavidin agarose beads (Millipore) overnight, rotating in the dark at 4 °C. The next day, beads were washed with cold lysis buffer, and washed once in 1 M NaCl. At this point, each sample was resuspended in 100 μl of lysis buffer and divided as follows: 30 μl were used to test iPOND by western blot and were mixed with LB sample buffer as their relative inputs. The remaining 70 μl and relative inputs were processed for RNA isolation. First, crosslink reversal was performed in proteinase K buffer. Finally, samples were resuspended in Trizol and RNA was isolated to test the expression of *lncREST* by RT-qPCR.

### In vivo RNA pulldown for mass spectrometry

In vivo *lncREST* pulldown was performed as previously described[55] with some modifications. DNA probes biotinylated at 3′ end targeting *lncREST* and LacZ were designed and purchased from IDT (see Supplementary Data 3 for sequences). Briefly, 22 × 10^7 HCT116 non-treated or treated with 2 mM HU overnight were washed with PBS and UV crosslinked with 265 nm UV light at 500 mJ/cm^2 on a UVC500 UV crosslinker, scraped and pellets were snap-frozen. The following day, pellets were quickly thawed, lysed (50 mM Tris-HCl, 10 mM EDTA, 0.5% SDS, supplemented with protease inhibitors and RNAse inhibitors), and sonicated in a Bioruptor device set on high for 15 cycles (30″ ON/45″ OFF) to obtain fragments of 100-500 bp. Sonicated lysates were centrifuged at high speed to remove debris and supernatant was pre-cleared with Dynabeads MyOne streptavidin C1 (Invitrogen) for 30 min, and beads were discarded. Pre-cleared lysates were diluted with hybridization buffer (500 mM NaCl, 0.5% SDS, 50 mM Tris-HCl pH7, 1 mM EDTA, 15% formamide, protease inhibitors and RNAsin) in a ratio 2 ml/ml of lysate and one percent and 10 percent inputs for RNA and proteins, respectively, were put aside. Lysate were combined with 200 pmoles of probes/ml of lysate for 4 hrs at 37 °C, and then leave overnight at RT. The next day, 200ul of washed dynabeads/ml of lysate were added to the hybridized samples for 4 h at 37 °C. Beads were washed for 5 times, 5 min each wash with wash buffer (2X SSC, 0.5%SDS, supplemented with protease and RNAse inhibitors). Finally, beads were resuspended in elution buffer (20 mM Tris HCl pH8, 2 mM MgCl2, 0.05% sodium lauryl sulfate, 0.5 mM DTT, Benzonase −125U for 200 ×10^6 cells, RNaseA) and incubated with agitation for 2–3 h before separating the elute from the beads. The remaining beads were resuspended in 50 mM Ammonium bicarbonate pH 8. Both elutes and beads were sent for mass spectrometry. We analyzed one LacZ sample and *n* = 2 lncREST pulldowns. Beads were washed at least five times with 100 μl of 50 mM ammonium bicarbonate, then 5 μl (200 ng/ μl) of modified sequencing-grade trypsin (Promega, Madison, WI) was spiked in and the samples were placed in a 37 °C room overnight. The samples were then centrifuged or placed on a magnetic plate if magnetic beads were used and the liquid removed. The extracts were then dried in a speed-vac (1 h). All samples were then re-suspended in 50 μl of HPLC solvent A (2.5% acetonitrile, 0.1% formic acid) and desalted by STAGE tip[56]. On the day of analysis, the samples were reconstituted in 10 μl of HPLC solvent A. A nano-scale reverse-phase HPLC capillary column was created by packing 2.6 μm C18 spherical silica beads into

a fused silica capillary (100 μm inner diameter x 30 cm length) with a flame-drawn tip[57]. After equilibrating the column each sample was loaded via a Famos auto sampler (LC Packings, San Francisco CA) onto the column. A gradient was formed and peptides were eluted with increasing concentrations of solvent B (97.5% acetonitrile, 0.1% formic acid). As peptides eluted, they were subjected to electrospray ionization and then entered into a Velos Orbitrap Elite ion-trap mass spectrometer (Thermo Fisher Scientific, Waltham, MA). Peptides were detected, isolated, and fragmented to produce a tandem mass spectrum of specific fragment ions for each peptide. Peptide sequences (and hence protein identity) were determined by matching protein databases with the acquired fragmentation pattern by the software program, Sequest (Thermo Fisher Scientific, Waltham, MA). All databases include a reversed version of all the sequences and the data was filtered to between a one and two percent peptide false discovery rate. To retrieve *lncREST* protein interactors, we considered only proteins identified by ≥1 peptide only for proteins exclusively identified in the *lncREST* probe samples and not in LacZ control. For proteins that were present in both *lncREST* pulldown and LacZ sample, we first considered proteins which ratios between avg intensity in *lncREST* probe samples compared to LacZ was ≥3. For the shortlisted proteins, we further discarded those with a ratio of unique peptides *lncREST*/LacZ was <2. Finally, we subjected the list of candidate proteins to databases reporting common mass spectrometry contaminants, like Mascot database on Matrix Science (www.matrixscience.com) and[58], filtering out unspecific proteins.

### In vitro RNA pulldown

Full length *lncREST* was cloned from from a pcDNA3.1(+) plasmid to a pT7T3D-PacI plasmid (Source Bioscience), which was then was linearized by restriction digestion with NotI (for the sense sequence) or EcoRI (antisense sequence). *LncREST* was in vitro transcribed with T7 RNA polymerase (Promega) (sense and fragments) or T3 polymerase (Promega) (antisense) and was labeled with biotin using Biotin RNA labeling mix (Roche), according to the manufacturer's instructions. To obtain *lncREST* fragments, forward oligos with T7 tail on the 3′ end, and Rev primers at different positions on *lncREST* sequence were designed for in vitro transcription. Primers sequences are listed in Supplementary Data 3. In vitro RNA pulldown was performed as previously described[59]. The recovered proteins were subjected to western blot.

### RNA immunoprecipitation (RIP)

For NCL immunoprecipitation, 3 × 10^7 cells non-treated or treated with HU 3.5 mM for 6 hrs were UV crosslinked 265 nm UV light at 500 mJ/cm^2 on a UVC500 UV crosslinker. Snap-frozen pellets were resuspended in 1 ml of lysis buffer (10 mM Hepes pH7.4, 100 mM KCl, 5 mM MgCl_2, 0.5% NP-40, 1 mM DTT, supplemented with protease inhibitors and RNAsin) using a Douncer, and sonicated 15 min in a Bioruptor device set on high (10″ ON/40″ OFF). Lysates were cleared by centrifugation at 9000 x g for 10 min. After pre-clearing with Dynabeads protein A (Thermo scientific), 1% inputs were collected for protein and RNA analysis, and supernatants were incubated with 5 μg of NCL or IgG isotype control for 4 h at 4 °C under rotation. Then, 100 μl of Dynabeads protein A were added and incubated for 2 h, then beads were washed 5 times with NT2 buffer (50 mM TrisHCl pH 7.5, 150 mM NaCl, 1 mM MgCl, 0.05% NP-40, supplemented with protease inhibitors and RNasin), and finally resuspended in 100 μl of NT2 buffer.

A total of 20% washed beads solution was resuspended in LB sample buffer to test NCL, pulldown by western blot, and the remaining 80% was brought to 200 μl volume in Proteinase K/NT2 or RIP buffer (NT2 or RIP buffer with 1% SDS, 2.4 μg/μl proteinase K) and de-crosslinked at 55 °C for 30 min. Finally, 700 μl of TRIzol (Invitrogen) were added and RNA was isolated. Following cDNA synthesis, samples

were analyzed by RT-qPCR to identify *lncREST* enrichment over input samples.

## CO-IP

Co-IP experiments were performed as previously described with some modifications[60]. Briefly, asynchronous cells were resuspended in nuclei isolation buffer (NIB) 1X (260 mM sucrose, 8 mM Tris-HCl pH 7.4, 4 mM $MgCl_2$, 0.8% Triton X-100) for 30 min on ice, mixing every 5 minutes, and nuclear pellets were collected by centrifugation (1125 g for 15 min). Nuclear pellets were washed with NIB 1X without Triton and without sucrose, and lysed with Buffer A (20 mM TrisHCl pH 7.5, 250 mM NaCl, 5 mM $MgCl_2$, 5% glycerol, 0.1 mM EDTA, 0.3% NP-40, 1 mM DTT, 1 mM $CaCl_2$, 20 mM MG132, supplemented with protease and phosphatase inhibitors) using a Douncer, incubating for 30 min at 4 °C under rotation, vortexing every 5-10 min. Lysates were then diluted by adding equal amount of Buffer B (Buffer A without NaCl and NP-40) and centrifuged to remove nuclear membranes at 9000 x g, 4 °C. Lysates were then quantified by BCA assay (Pierce) to equalize protein amounts for the different samples and divided in two (one for the specific antibody, and IgG isotype control). After pre-clearing with washed Dynabeads proteins A/G (Invitrogen), beads were discarded and lysates (1 mg) were incubated with 5 μg of specific antibodies and control IgG, respectively, for 4 h at 4 °C under rotation. Lysates were then incubated with Dynabeads proteins A/G for 1 hr at 4 °C and beads were washed for 5 times, 5 min at 4 °C in Buffer A/B. Finally, beads were resuspended in LB sample buffer and analyzed by western blot.

## Immunofluorescence of mitotic cells

Immunofluorescence of mitotic cells was performed according to[61] with some modifications. Cells were seeded on coverslips ($2 \times 10^5$ cells in each well of a 6-well plate) and transfected with LNA GapmeRs targeting *lncREST* or siRNAs against *NCL* using RNAiMax, according to the manufacturers' protocol. 48 h post-transfection, cells were treated with 500 μM HU for 3 h. Following treatment, cells were washed with PBS and incubated with ice-cold culture medium for 10 min on ice and fixed with cold methanol for 3 min at -20 °C. Coverslips were then washed with rehydration buffer (CBS) (137 mM NaCl, 5 mM KCl, 1.1 mM $Na_2HPO_4$, 4 mM EGTA, 4 mM $MgCl_2$, 10 mM PIPES, 10 mM sucrose, adjusted to pH 6) twice, 5 min each wash, followed by 5 min PBS wash. Cells were then washed two times for 5 min with PBST-0.01% Triton X-100, and incubated with blocking solution (PBST with 1% BSA) for 1 h at RT to block unspecific antibody binding. Then, slides were incubated with primary antibodies p-CENP-A (Cell Signaling) 1:100, α-tubulin (Millipore) 1 μg/mL diluted in blocking solution for 1 hr at 4 °C. Slides were then washed 3 times with PBST and incubated with AlexaFluor conjugated secondary antibodies (Invitrogen) diluted in PBST at 1:500 dilution, for 1 h at RT. After three PBST washes, coverslips were mounted with nuclear blue (Invitrogen). Mitosis were analyzed on a Zeiss Axio Imager M1 at 63X magnification.

## Cell proliferation assay and apoptosis

For proliferation assay, 1000 cells/well were plated in 96-well plates and cell proliferation was measured over 3 or 4 days with a CellTiter96 Aqueous Non-Radioactive Cell Proliferation Assay (MTS) Kit (Promega). Absorbance at 490 nm was measured by spectrophotometry using the SPECTROstar Nano 96-well plate reader (BMG Labtech). Apoptosis was assayed by Annexin V and 7-AAD staining using the Apoptosis Detection Kit I (BD Biosciences) and FACS Canto flow cytometer, following the manufacturer's recommendations. BD FACSDiva v 8.0.1 was used to collect data, and data were analyzed with BD FlowJo v10.

## Xenograft experiments

Animal studies were performed in accordance with our Ethical permit (006-20), reviewed and approved by the Comité de Ética para la

experimentación animal of the University of Navarra. Mice were kept under the following housing conditions: 12 h light/12 hrs dark cycle, at 18-23 °C and 40-60% humidity. $3.5 \times 10^6$ HCT116 cells transfected with *lncREST* targeting or CTRL LNA GapmeRs, or *lncREST* KO and wt cells were re-suspended in 100 μl of complete medium and mixed with Matrigel Matrix (Corning) in a ratio of 1:1. The resultant mix was injected subcutaneously in the flank of 6–7-weeks-old female BALB/cA-Rag2$^{-/-}$γc$^{-/-}$ immunodeficient mice ($n = 4$ per condition for *lncREST* KD, $n = 6$ per condition in *lncREST* KO experiment).

## Reporting summary

Further information on research design is available in the Nature Portfolio Reporting Summary linked to this article.

## Data availability

The RNA-seq data reported in this study have been deposited in NCBI's Gene Expression Omnibus (GEO) repository under the GEO series accession number GSE229870. The mass spectrometry proteomics data have been deposited to the ProteomeXchange Consortium via the PRIDE partner repository with the dataset identifier PXD043367. Source data are provided with this paper. ChIP data used to identify peaks in lncREST promoter can be found in ChIP atlas (http://chip-atlas.org). ChIP-seq data from HCT116 cells treated with DNA damaging agent 5-FU were obtained from NCBI's GEO database under GEO series accession number GSE58507. Repli-seq data on HCT116 cells were obtained from[18] under GEO series accession number GSE137764. Source data are provided with this paper.

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

## Acknowledgements

This work was supported by grants to M.H.: Fondo Europeo de Desarrollo Regional/Ministerio de Ciencia, Innovación y Universidades — Agencia Estatal de Investigación grant BFU2017–82773-P, European Research Council Consolidator grant 771425 and Worldwide Cancer Research grant 20-0204. M.M is funded by a MSCA fellowship (HORIZON-MSCA-2021-PF-01: 101066499).

## Author contributions

M.H. and L.S. conceived the study. L.S. designed, performed, and analyzed most of the experiments. JM.F.J. performed most of the bioinformatic analyzes. J.G. assisted in the experimental work, generated the stable dCAS9/KRAB cells and performed the in vitro pull-downs. A.R. performed the experiments for the identification of RNA on the replication fork, immunofluorescence of CRISPR KO and CRISPRi cells, and RIP experiments in collaboration with L.S. E.G. analyzed RNA-seq from chromatin samples and synchronized cells. M.M. performed treatments, WBs, and qPCR shown in Supplementary Figs. 3 and 6, contributed to the revision, and editing of the final manuscript. A.M.M. performed the cell fractionation experiments in collaboration with L.S., a subset of immunofluorescence experiments, and the Co-IP experiments. M.H. and L.S. wrote the manuscript with editorial help from J.M.F.J., MM and A.M.

## Competing interests

The authors declare no competing interests.
