## [Peer Review File · Nature Communications]

The chromatin-associated IncREST ensures effective replication stress response by promoting the assembly of fork signaling factorsREVIEWER COMMENTS

Reviewer #1 (Remarks to the Author):

In this manuscript Statello et al. identified new long noncoding RNA (lncRNA) whose function is relevant to DNA replication and genome stability. In particular, the authors identified lncRNAs localized on the chromatin whose expression is responsive to replication stress treatment. To induce replication stress, the authors treated the HCT116 colon cancer cells with hydroxyurea (HU). They identified lncRNA called lncREST, a chromatin enriched lncRNA that is upregulated upon HU and whose expression is also enriched in S stage of the cell cycle. Mechanistically lncREST interacts with nucleolin (NCL), a very abundant RNA binding protein. lncREST helps NCL to interact with replication protein RPA, and depletion of lncREST leads to defects in replication fork progression, accumulation of DNA damage and reduced NCL-RPA interaction.

Overall this is an interesting study but some conclusions and claims need to be supported with additional experiments before publication. My main concern is that lncREST depleted cells already have proliferation defects even in the absence of DNA damage, they have increased apoptosis, high levels of DNA damage and increased fork speed. These data suggest that lncREST depleted cells are experiencing a lot of stress even without replication stress or DNA damage.

Furthermore, the authors need to strengthen their data on replication stress (RS) and show that the defects they are seeing are not indirect DNA damage caused by HU. This for example could be done by treating the cells with additional drug known to induce RS, such as aphidicolin and showed main finding in hTERT-RPE1 cells. The authors need to be aware that although double thymidine block is widely used to synchronize the cells, it is known to induce DNA damage. Thus, upregulation of lncREST in S phase could be caused by thymidine-induced DNA damage. This needs to be demonstrated to make sure that lncRNA expression is indeed cell cycle regulated.

The manuscript is easy to read and the figures are nicely represented. Some of the figure legends don't match the text, so please correct that.

The methods are sound, and expected standards are met. Some part of the manuscript require more experiments to make sure the data are reproducible and supporting the authors claims.

Recommendations for the authors:

The points to be addressed before proceeding to final publication are listed below and divided in Major and Minor corrections. We hope the authors find them useful.

Major corrections:

- My main concern is that lncREST depleted cells even without any treatment with DNA damaging agents have proliferation defects (Fig 7), show signs of genome instability (Fig 3D), have increased apoptosis, decreased levels of RPA and increased fork speed (Supp Fig 4). The authors should show comment on that. They should also show cell cycle profile of lncREST depleted cells as it is possible that they are stuck in G1 phase of the cells. It is possible that lncREST have a role in replication fork progression in the normal conditions (in the absence of RS).
- Why did the authors choose to use 1mM HU treatment for 8hr without any release for RS? Most of the people will use 2mM HU treatment (16hr) followed by the release.
- It is not clear to me why the authors used so many different concentration of HU throughout the manuscript with different timing? Different concentration of DNA damaging agents and incubation time has a different effect on DNA repair and DNA damage signaling.
- To confirm the role of lncREST in RS, the authors should use another drug used to activate RS such as aphidoclin (APHI, low dose 400uM, 24hr). Use of APHI will strengthen their data that lncREST is

upregulated by RS and not caused by indirect DNA damage

- RS results in several phenotypes such as DNA damage in prometaphase, ultrafine anaphase bridges (UFBs) and 53BP1-positive nuclear bodies in G1 cells. Since it is difficult to distinguish between DNA damage and RS, the authors should demonstrate that in addition to DNA fibre assay, IncREST depleted cells also have DAPI negative anaphase UFBs (coated with RPA or PICH or BLM) or 53BP1 bodies in G1 phase (CyA negative) or increased H2AX foci in prometaphase (not interphase) cells. These assays correlate well with RS, and are easy to perform by IF. These experiments will strengthen their data and will demonstrate that IncREST is involved in RS and not activated indirectly by DNA damage. Indeed, different DNA damaging agents induce the expression of IncREST (Fig 2E), so how confident are the authors that this response is really RS specific?
- Since replication stress leads to chromosome segregation errors during mitosis, the authors need to demonstrate that IncREST depleted cells have increased rate of anaphase cells with lagging chromosomes or chromatin bridges (especially in the light of the presented data that IncREST depleted cells have mitotic defects, Fig 7).
- Fig 1A, B: please show the cell cycle profile upon 1mM HU, 8h treatment. That would show how cell cycle is disrupted upon HU.
- Fig 1C: the authors should show the overlap of IncRNAs in total and chromatin fraction (eg venn diagram).
- Fig 1D: did the authors explore IncREST expression in other human cell lines where similar methods have been to study IncRNA expression during cell cycle synchronization: hTERT-RPE1 cells (PMID: 32402276) and U2OS (PMID: 33108271)?
- Although the cell cycle has been studied through widely used thymidine induction synchrony, double thymidine block is known to lead to DNA damage (PMID: 16672000, PMID: 33052073) and chromosomal aberrations (PMID: 5929561). Thus these data needs to be interpreted with caution. Indeed In the last few year the cell cycle community is using a palbociclib, a CDK4/6 inhibitor that can arrest cells in G1 phase leading to minimal DNA damage (PMID: 33052073, PMID: 35037284). Therefore the authors should show that double thymidine block used to synchronize HCT116 is leading to any DNA damage (eg. H2AX foci) at the same point where IncREST is induced (Fig 1E, Fig 2D)
- Fig 2A: please show the tracks from untreated and HU treated cells (total and chromatin for IncREST).
- Supp Fig 2A: is IncREST chromatin bound also in untreated cells? Did the authors compared the expression of IncREST by cell fractionation in untreated and HU treated cells? Please clarify the Supp Fig 2A
- The authors should be consistent in representing the data in the main and supplementary figures. Please include individual values in all the graphs throughout the manuscript.
- Since RPE1 cells, a normal diploid cells, are widely used to study cell cycle and have a higher levels of IncREST (based on Supp Fig 2G) it would be worth of testing test whether treatment of RPE1 cells with HU and APhi leads to upregulation of IncREST also in RPE1 cells. These two drugs are commonly used to induce RS in RPE1 cells.
- Fig 2D: the authors should show that no DNA damage is caused by double thymidine block to exclude that upregulation of IncREST in early S is not due to DNA damage, but rather its expression is cell cycle stage specific. The authors could also include qPCR for S phase genes such as cyclin E1 or D1 that are expressed in early S phase. The authors show provide H2AX staining for each time point to exclude DNA damage. This is an important experiment as the authors are showing that IncREST can be upregulated upon treatment of cells with different DNA damaging agents (Fig 2E).
- Fig 3: Comet assay and micronuclei are not usually considered markers of RS, but rather markers of DNA damage. To show RS, the authors should perform IF for above mentioned markers.
- Fig. 3C: is the HU treatment ON? Why this treatment is different from the initial treatment (8hr)? Please include also untreated control cells upon IncREST depletion. The same is valid for H2AX foci in Fig 3D.
- Fig 3C: The micronuclei look slightly bigger that they should be. Please zoom into one cell for each condition. Micronuclei are usually around 2-2.5 μm in size, but these nuclear structures look bigger. Is it possible that some of the micronuclei scored in this assay might be nuclear blebbing caused by apoptosis? Can the authors stain micronuclei with lamin B1 and H2AX. I am expecting that these

micronuclei will have no lamin B1 staining and high levels of DNA damage. Please show the graph also for untreated cells upon IncREST depletion to have an idea of basal level of micronuclei formation without HU. I suggest to show untreated and HU treated cells for every IF image, as done with CRISPR-mediated depletion and H2AX staining in Supp Fig 3. Please specify how many experiments were performed for that staining (Supp Fig 3C-D).

- Do CRISPRi and CRISPR KO cells for IncREST have higher rate of micronuclei upon HU? The same question is for defects in ATR/CHK1 signalling. In addition to IF, defects in ATR/CHK1 signalling are usually shown by western blot. Thus, the authors should confirm defects in ATR/CHK1 signalling by using antibodies for total ATR, ATR phospho, CHK1 total, CHK1 phospho and RPA phospho pRPA 32 (especially if these cells have a ssDNA gaps) upon HU (and APhiI) after LNA, CRISPRi and/or CRISPR KO cells for IncREST. Alternatively DNA fiber assay should be performed with one of the CRISPR methods.
- Fig 4A: the authors should clarify why the 50uM Hu was used, and why HU was added during the second pulse and not after Cldu?
- Fig 4C: the authors should demonstrate the same graph but without S1 endonuclease; that is a standard for this experiment.
- Fig 5: Why 3mM HU was used for iPOND assay. It is not clear why different HU concentration are used throughout the manuscript.
- The authors should show quantitative comparison of HU-treated cells in control (luc) and IncREST pulldown (either by SILAC or TMT). Showing the graph with peptide intensities is not the standard in the proteomics as it is possible that NCL is also present in the control (luc) cells. Please show the data as volcano in HU treated cells to indicate significant interactors.
- The authors should show NCL enrichment in Fig 5B by western. Is NCL recruited to chromatin upon HU where it could interact with IncREST?
- Fig 6D: What is the pattern of NCL staining in untreated cell (no IncREST depletion). It is important to show NCL staining in untreated and treated cells (HU).
- Fig 6E: is this untreated or HU treated cells? Please indicate in the Figure. What is the level of RPA32 in untreated cells? From Fig 6G it is clear that IncREST depleted cells without any damage have less RPA in the input. These data suggest that IncREST depleted cells are experiencing defects in replication even in the absence of any RS.
- Fig 7: What is the cell cycle profile of IncREST depleted cells? It is not surprising to find less interaction between RPA and NCL upon HU since IncREST depleted cells have lower RPA levels in untreated cells, and are proliferating less.
- Is RS occurring in nucleoli upon HU in IncREST depleted cells? Do authors find TOPBP1 and Tracle in their proteomics experiment (see PMID: 34100862)?
- If IncREST is involved in RS response, that IncREST depleted cells should be more sensitive to ATR, Wee1 or CHK1 inhibitors. Did the authors tested the sensitivity of these cells?
- Fig 7A; to show the link to CIN, the authors need to demonstrate that IncREST depleted cells have anaphase cells with lagging chromosomes or chromatin bridges. Please check the figure legends, as the text is not corresponding to the images.
- Fig 7B: The quality of images showing mitotic defects needs to be improved. How did the authors define misaligned chromosomes? It is not clear from images in Fig 7 that CENPA signal is outside of the metaphase plate upon IncREST depletion. In addition, the cells seems to have problems with microtubules (increased microtubule stability was shown to be associated with RS). It would be interesting to show whether these cells have anaphase cells with lagging chromosomes or chromatin bridges (it is possible that the cells will arrest in metaphase due to spindle defects). Scale bar is missing.
- What is the difference in mitotic defects shown in Supp Fig 7B and in Fig 7B?
- Based on Drows et al, RS is associated with eight signatures suggesting it is a major source of CIN. In addition to CX9 signature, there are other signatures (eg CX13, CX8, CX11) (line 315) that did not correlate with low RS. What could be the reason?
- Sup Fig 7B- scale bar is missing. It seems to me these cells have multipolar spindles based on tubulin staining, is that the case?

Minor corrections:

- Fig 6F: there is a mistake in the image, the staining is for NCL but quantification is showing RPA32. Please correct.
- What is Fig 6B showing? Please indicate the western blot in the image.
- Include catalogue number of the drugs and reagents used in this study (line 425).
- FOXM1 is a transcription factor with a known role to regulate expression of genes in G2/M, therefore I would tune the sentence (line 111) that FOXM1 has a well-known role in RS (based on one paper cited). This should be corrected in the text.
- Did the authors try smRNA fish using Stellaris FISH probes as they did in the past ?
- Line 551: should be HU not "hu".
- Line 726: should be RNA-seq not RNAseq
- Include page numbers in the manuscript.

Reviewer #2 (Remarks to the Author):

Statello et al. describes the involvement of IncREST in the cellular response to replication stress. The paper is well-written, the data and methods are comprehensive, and the conclusions are, for the most part, supported by the presented data. The identification of IncREST and its association with NCL is very convincing, as well as the impact of IncREST for genome stability and tumor growth. Mechanistically, the function of IncREST is more difficult to assess, as the involvement of its binding partner NCL itself in replication stress is not established. Nevertheless, the results are novel, interesting and important. I feel the manuscript is appropriate for the broad readership of Nature Communications but there are a few major issues that I believe need to be clarified before publication.

Here are my detailed comments:

Figure 1

1A: Add total protein for ATR, p53 and H2AX in WB

1B-C: Ok

1D: Is the figures showing total or chromatin-associated RNA, or both?

1E-F: Ok

1G: Include color explanation in figure or legend

Figure S1A-C: Ok

Figure 2

The figures could be re-organized to better fit their appearance in the text.

2A: Ok

2B: Typo = cytoplasm should be spelled cytoplasm

2C: Are these dots of IncREST corresponding to sites of replication stress? I would be interesting to combine the FISH staining +/- HU with staining of RPA and NCL

2D-G: Ok

Figures S2A-H: OK

Figure 3

3A: Ok

3B-C: Show comet assay and micronuclei number also in no-HU conditions

3D: The reduction of gH2AX is strange and cannot solely be explained by an attenuated activation of ATR. Is ATM also impaired following loss of IncREST and/or siNCL?

3E: It would be good to assess levels of gH2AX, pATR and pATM (as well as total counterparts) after LNA IncREST employing western blot (WB). Include both NT and HU conditions. Assessment of protein levels by WB and also in no-HU conditions is important through the manuscript.

Figures S3A-B: Ok

Figures S3C: Are the stable KO/CRISPRi cells also showing signs of DNA breaks and/or micronuclei?
Figure S3D-J: Ok

Figure 4
Is loss of IncREST affecting cell cycle distribution?

Figure 5
5A: Ok
5B: Is NCL among the proteins obtained by iPOND? The blot for RAD51 is dirty and bands cannot be assessed properly.
5C-F: Ok
Figures S5A-D: Ok

Figure 6
More controls linking NCL and replication stress would strengthen the study.
6A: Ok
6B: The NCL label in the top blot is missing.
6C: To assess how much of the chromatin-bound NCL that reflects binding to stalled replication forks, a WB should be included showing NCL binding to chromatin +/- HU. Also include LNA IncREST in both +/- HU in this blot.
6D: It would be good to include evidence that NCL changes localization upon replication stress (i.e., +/- HU), to more firmly establish that IncREST facilitates this movement. Now only +HU is shown - At text row 275, the authors write "p-ATR, recruitment that we have found to be affected". This has not been shown, only reduced levels of pATR. Either examine recruitment using IP or IF of pATR followed by co-precipitation or co-staining of RPA, or re-phrase this sentence.

6E: It would be good to include chromatin-association of RPA32 in IncREST-depleted cells also in no-HU condition (similar blot as S6A)

What is the reason the reduced levels of RPA in chromatin following knockdown of IncREST? Impaired generation of ssDNA or reduced binding of RPA to ssDNA? This is important to explore, as the reduced levels of RPA in chromatin likely underlies the attenuated pATR signal.

6F: I think the NCL legend on top of middle image should be replaced for RPA32
6G: Ok
S6A: This WB is very informative as it includes both NT and HU conditions.
S6B: Ok
S6C-D: Show a WB of chromatin-associated RPA32 after siNCL +/- HU and in a similar style as the blot in S6A. Also blot for pATR and gH2AX, and total counterparts.
S6E-F: Ok

Figure 7
There seem to be a mixup with main, suppl figures and legends here.

Is expression of IncREST showing any correlation to patient survival? If downregulation of IncREST kills cancer cells one would think that tumor cells would avoid this, or?

A model figure would be good to include.

The authors propose that loss of IncREST results in replication stress. However, the signs of replication stress, i.e., elevated RPA in chromatin, elevated pATR and gH2AX are not seen after LNA IncREST, but instead a reduction of these factors. What could be the mechanistic explanation of this? This should be discussed in the manuscript.

Reviewer #3 (Remarks to the Author):

In this manuscript, Statello et al seek to identify lncRNAs that are induced by replication stress, accumulate in chromatin, and might participate in the cellular response to stalled forks. They focus on lncREST, which harbors regulatory motifs for transcription factors involved in the replication stress response, including a p53 response element. The authors use LNA, CRISPR gene deletion KO and CRISPRi as three alternative LOF approaches and observe mitotic defects, increased apoptosis, and, as a result, impaired proliferation. Overall, this manuscript identifies a new lncRNA (see point 1 below) and identifies cellular phenotypes associated with perturbation of various functional elements in its locus. However, revisions are requested to address the potential cis regulatory function of the lncRNA locus (point 2) and the potential functional but non-specific interactions with protein factors (point 3).

- 1) The authors refer to this lncRNA as a novel and uncharacterized lncRNA, giving it the name lncREST. However, the same lncRNA was recently described and named PTL5 (p53-regulated tumor-suppressive lncRNA) by another study (Mitra, Adams, and Eischen, Cancer Res Comm 2023). The authors should update their manuscript accordingly to reflect the published name.
- 2) It is very strange that the authors provide expression data for NDUFAF6 but not TP53INP1 RNA levels in Fig. 2, especially given that the p53 peak is shared between the two genes and given the effects of LNA and CRISPR KO/CRISPRi on TP53INP1 levels, discussed in Supplementary Figures. The effects of lncREST LOF perturbations on TP53INP1 are concerning and suggest a role for this lncRNA locus in regulating TP53INP1 expression, which in turn may be the driver of the S-phase phenotypes. This should be addressed in a revisions. The opposite effects of lncREST LOF perturbations on TP53INP1 may simply reflect the simultaneous presence of stimulatory and inhibitory elements in the locus that are perturbed differentially by the perturbations, as recently seen for other lncRNAs (ie lncRNA Meteor, Cell Reports 2023).
- 3) The protein interaction data (including NCL) is strong and interesting but it would be important to determine whether other lncRNAs identified in Fig. 1 as S-phase/replication fork-specific and chromatin-bound are also enriched for these factors. It is conceivable that this might reflect a general protein/nascent RNA aggregation feature in the chromatin at stalled replication forks analogous to SAF-A (Creamer et al Mol Cell 2021.)

Reviewer #1 (Remarks to the Author):

In this manuscript Statello et al. identified new long noncoding RNA (lncRNA) whose function is relevant to DNA replication and genome stability. In particular, the authors identified lncRNAs localized on the chromatin whose expression is responsive to replication stress treatment. To induce replication stress, the authors treated the HCT116 colon cancer cells with hydroxyurea (HU). They identified lncRNA called lncREST, a chromatin enriched lncRNA that is upregulated upon HU and whose expression is also enriched in S stage of the cell cycle. Mechanistically lncREST interacts with nucleolin (NCL), a very abundant RNA binding protein. lncREST helps NCL to interact with replication protein RPA, and depletion of lncREST leads to defects in replication fork progression, accumulation of DNA damage and reduced NCL-RPA interaction.

Overall this is an interesting study but some conclusions and claims need to be supported with additional experiments before publication. My main concern is that lncREST depleted cells already have proliferation defects even in the absence of DNA damage, they have increased apoptosis, high levels of DNA damage and increased fork speed. These data suggest that lncREST depleted cells are experiencing a lot of stress even without replication stress or DNA damage.

Furthermore, the authors need to strengthen their data on replication stress (RS) and show that the defects they are seeing are not indirect DNA damage caused by HU. This for example could be done by treating the cells with additional drug known to induce RS, such as aphidicolin and showed main finding in hTERT-RPE1 cells. The authors need to be aware that although double thymidine block is widely used to synchronize the cells, it is known to induce DNA damage. Thus, upregulation of lncREST in S phase could be caused by thymidine-induced DNA damage. This needs to be demonstrated to make sure that lncRNA expression is indeed cell cycle regulated.

The manuscript is easy to read and the figures are nicely represented. Some of the figure legends don't match the text, so please correct that.

The methods are sound, and expected standards are met. Some part of the manuscript require more experiments to make sure the data are reproducible and supporting the authors claims.

We thank the reviewer for finding our manuscript of interest. We have tried to address the points raised to strengthen our claims.

Recommendations for the authors:

The points to be addressed before proceeding to final publication are listed below and divided in Major and Minor corrections. We hope the authors find them useful.

Major corrections:

- My main concern is that IncREST depleted cells even without any treatment with DNA damaging agents have proliferation defects (Fig 7), show signs of genome instability (Fig 3D), have increased apoptosis, decreased levels of RPA and increased fork speed (Supp Fig 4). The authors should show comment on that. They should also show cell cycle profile of IncREST depleted cells as it is possible that they are stuck in G1 phase of the cells. It is possible that IncREST have a role in replication fork progression in the normal conditions (in the absence of RS).

We understand the reviewer's concerns about the effects of *IncREST* in NT cells. It has to be kept in mind that HCT116, as highly proliferating cancer cells, are subjected to a basal level of replication stress. Indeed, as we can see from the WBs in Figure 1, there is a basal expression of pATR and γH2AX in NT samples, which could be the reason why we observe some phenotypes also in NT cells.

We have performed cell cycle analysis of cells *IncREST*-depleted cells in non-treated condition and after treatment with HU at the same conditions used for the initial RNA sequencing (1mM 8hrs), and APH 40 uM for 24h (**Figure R1**). We found that while in NT cells there is no significant effect on the cell cycle, after both treatments *IncREST* depleted cells slightly accumulate in G2 phase, which is consistent with the observed phenotypes. *IncREST* KD enables replication of damaged DNA (as shown by the fiber assays) and elicits progression of cells through the S phase, leading to G2 arrest resulting from persisting DNA damage. We have added these data to the revised version of the manuscript in the **Figure 3B**.

Figure R1. Cell cycle profile of HCT116 cells untreated or upon RS treatments (1mM HU for 8hrs, 40uM APH 24h) and transfected with the indicated LNAs.

- Why did the authors choose to use 1mM HU treatment for 8hr without any release for RS? Most of the people will use 2mM HU treatment (16hr) followed by the release.

We thank the reviewer for raising this important point that we didn't sufficiently explain in the manuscript. The election of the time and concentration of the treatment was a careful and conscient decision. Our initial goal was to identify lncRNAs specifically activated in response to replication stress, before the DNA damaging and repair pathways are fully activated, in order to detect new possible RNA mediators of DNA replication. Separating these pathways is not an easy task, given that repair occurs during replication to grant the correct duplication of cellular genomes. Moreover, some lncRNAs are typically activated during DNA damage. It is known that HU affects replication in a dose-dependent manner, and that high levels of prolonged HU (about 24hrs) exposure causes activation of the ATM S-phase checkpoint kinase (Snyder et al., 2009) (Shechter et al., 2004), besides the ATR pathway. In establishing the concentration and time of

our HU treatment for RNAseq analysis, we aimed at 1) minimizing the overlap between the replication stress and DNA damage/repair pathways; 2) cause an effect that is reversible, in order to avoid cell cycle exit. 1mM of HU treatment for 8 hrs allowed us to see an activation of the replication stress pathway without observing a strong activation of the ATM pathway (**Figure 1**, (Goder et al., 2018)), an effect that is rapidly reversed by removing the drug and letting cells recover for at least 3 hours in normal medium.

We believe that performing recovery after HU treatment would not be a better strategy, as the return of cells to their normal status following the recovery would not allow us identify the direct transcriptional changes as intended.

- It is not clear to me why the authors used so many different concentration of HU throughout the manuscript with different timing? Different concentration of DNA damaging agents and incubation time has a different effect on DNA repair and DNA damage signaling.

We agree with the reviewer that different HU treatment conditions have different effects on dynamics of DNA damage signaling pathways. The different concentrations are typically used for the different assays, to allow better quantification of the phenotypes. This is the reason why we have applied different conditions to evaluate the extent of *IncREST* mechanistic function in these processes. This approach allowed us to identify several aspects of the RS response that are regulated by *IncREST*. For example, 1mM HU for 8hrs to o.n., and 2mM for 5hrs have shown to have similar effects on *IncREST* activation as well as on the replication stress markers pATR and γ H2A.X, being known that these markers are activated already at very early time points following replication stress induction, and accumulate over time (**Figure3**).

On the other hand, it is established that doses at least 10 fold lower than 1mM strongly impair fork progression without causing S-phase arrest, therefore provide a better experimental window to appreciate some of the phenotypes under study (Wilhelm et al., 2014); (Nakatani et al., 2022)). This is why we have used micromolar concentrations of HU in the fiber assay experiments shown in Figure 4.

Finally, for NCL and RPA32 experiments shown in Figure 6, we have referred a publication studying the displacement of NCL in response to replication stress caused by HU at the same conditions, where *IncREST* is also activated ((Wang et al., 2021))

Despite different drug concentrations and times of treatments, we have verified in every experimental condition that cells are subjected to replication stress based on the protein markers and the fork phenotypes observed. Moreover, we believe that including multiple conditions reinforces the notion that *IncREST* is involved in establishing a correct response to replication stress.

- To confirm the role of *IncREST* in RS, the authors should use another drug used to activate RS such as aphidoclin (APHI, low dose 400uM, 24hr). Use of APHI will strengthen their data that *IncREST* is upregulated by RS and not caused by indirect DNA damage.

We thank the reviewer for this suggestion. We have now added *IncREST* expression data in HCT116 cells treated with low doses Aphidicolin, as suggested by the reviewer, to further support that replication stress induces *IncREST*, and is not just induced as a consequence of DNA damage. *IncREST* is upregulated by Aphidicolin, as we now show in **Figure 2E and R2A**. To verify

that at these conditions of APH treatment replication stress is induced, we have performed western blot on markers of replication stress in HCT116 treated with APH, similar to what we show for HCT116 treated with HU (WBs) in **Figure R2B**.

Figure R2. *IncREST* is induced by different stress agents. **A.** RT-qPCR analysis of *IncREST* expression in HCT116 cells treated with 2mM of hydroxyurea (HU) 8h, 40 μM Aphidicolin (APH), 24h, 15 μM of cisplatin (cis), 5 μM of doxorubicin (Dox), 20 μM of nutlin (Nut) o.n., 10 μM of camptotecin (CPT) for 8h, relative to normal condition (NT). **B.** Western blot (left) and relative quantification (right) of replication stress markers in HCT116 cells treated with aphidicolin (APH) compared to NT cells. N= 3 biological replicates. **p < 0.01, ***p < 0.001, ****P<0.0001 (mean ± SD, two-tailed unpaired t-test).

- RS results in several phenotypes such as DNA damage in prometaphase, ultrafine anaphase bridges (UFBs) and 53BP1-positive nuclear bodies in G1 cells. Since it is difficult to distinguish between DNA damage and RS, the authors should demonstrate that in addition to DNA fibre assay, *IncREST* depleted cells also have DAPI negative anaphase UFBs (coated with RPA or PICH or BLM) or 53BP1 bodies in G1 phase (CyA negative) or increased H2AX foci in prometaphase (not interphase) cells. These assays correlate well with RS, and are easy to perform by IF. These experiments will strengthen their data and will demonstrate that *IncREST* is involved in RS and not activated indirectly by DNA damage. Indeed, different DNA damaging agents induce the expression of *IncREST* (Fig 2E), so how confident are the authors that this response is really RS specific?

We appreciate the concern of the reviewer that *IncREST* may be involved in the DNA damage response rather than in replication stress. *IncREST* is indeed induced by different types of damaging agents. However, we have shown that the phenotypes associated to *IncREST* loss are consistent with a defective signaling at the first steps of replication stress response: unpaired recruitment of RPA and pATR, and decreased gamma-H2AX which leads to replication defects and increased damage. Moreover, we show that eventually the cells suffer mitotic defects, as described in Fig 7. Other secondary consequences correlating with these phenotypes are surely happening in these cells, although their study can be interesting, we respectfully believe that further detailed characterization goes beyond the scope of the manuscript.

- Since replication stress leads to chromosome segregation errors during mitosis, the authors need to demonstrate that *IncREST* depleted cells have increased rate of anaphase cells with lagging chromosomes or chromatin bridges (especially in the light of the presented data that *IncREST* depleted cells have mitotic defects, Fig 7).

As discussed in the previous comment, we respectfully believe that the data presented are sufficient to determine that the cells with lack of *IncREST* are dealing with defects in replication

stress signaling. While a more detailed study of further consequences of *IncREST* loss is of great interest, it is beyond the goals of the current study.

- Fig 1A, B: please show the cell cycle profile upon 1mM HU, 8h treatment. That would show how cell cycle is disrupted upon HU.

We have added to **Supplementary Fig. 1A** the cell cycle profile of HCT116 cells treated with 1mM HU for 8hrs, showing that cells are accumulating in G1/S phase as expected.

Figure R3. HU treatment induces G1 arrest. Cell cycle profile of HCT116 cells untreated or treated with 1mM HU for 8 hours.

- Fig 1C: the authors should show the overlap of lncRNAs in total and chromatin fraction (eg venn diagram).

We have added the overlap between chromatin and total fraction in the differentially expressed genes (lncRNAs and other gene types) detected by RNAseq analysis shown as a Venn diagram (**Supplementary Fig. 1** and **Figure R4**).

Figure R4. Venn diagram representing the differentially expressed genes (DEGs) up and down regulated in chromatin and total fraction. Numbers on the top: DE lncRNAs; numbers on the bottom: other DE gene types.

- Fig 1D: did the authors explore *IncREST* expression in other human cell lines where similar methods have been to study lncRNA expression during cell cycle synchronization: hTERT-RPE1 cells (PMID: 32402276) and U2OS (PMID: 33108271)?

As the reviewer requested, we have analyzed the expression of *IncREST* in the two studies ((Yildirim et al., 2020), (Hao et al., 2020)) for U2OS and RPE-1 synchronized cells. For U2OS cells, the different stages were obtained with a combination of double thymidine block followed by release at different time points (G1/S boundary, S-phase and G2-phase), and nocodazole treatment followed by mitotic shake-off (Mitotic cells and G1-phase). RPE-1 cells were

synchronized with mimosine. As we have observed in HCT116 cells, when U2OS cells are synchronized and released in different phases of cell cycle, we observe a peak of *IncREST* expression in the G1/early S time point, corresponding also to the peak of cyclin E. The experimental setting in RPE-1 cells was to perform gene expression in RPE-1 unsynchronized vs synchronized at different time points after the release. Here *IncREST* shows higher expression in the first three time points, representing different stages of the S-phase, and its expression significantly drops at later stages of release (T6-8, T8-10), as we have also observed in HCT116 cells and U2OS cells (**Figure R5**). These results confirm the induction of *IncREST* in S-phase in different cell types and with different synchronization methods, and have been added in the **Supplementary Fig. 2** of the revised manuscript.

Figure R5. *IncREST* is induced in early S phase. Graphs showing TPMs for *IncREST* and *Cyclin E* in U2OS and RPE1 cells, respectively, in cell synchronization followed by release at different cell cycle stages. Data are obtained from ((Yildirim et al., 2020), (Hao et al., 2020)).

- Although the cell cycle has been studied through widely used thymidine induction synchrony, double thymidine block is known to lead to DNA damage (PMID: 16672000, PMID: 33052073) and chromosomal aberrations (PMID: 5929561). Thus these data needs to be interpreted with caution. Indeed In the last few year the cell cycle community is using a palbociclib, a CDK4/6 inhibitor that can arrest cells in G1 phase leading to minimal DNA damage (PMID: 33052073, PMID: 35037284). Therefore the authors should show that double thymidine block used to synchronize HCT116 is leading to any DNA damage (eg. H2AX foci) at the same point where *IncREST* is induced (Fig 1E, Fig 2D).

As the reviewer suggested, we have performed cell cycle analysis and western blot for the experiment of HCT116 synchronization with double thymidine block followed by release, to identify the cell cycle markers and in parallel to the levels of DNA damage, based on γ H2AX protein detection. As shown in **Figure R6**, 1.5hrs after the release HCT116 are in early S phase, as shown by cell cycle analysis and confirmed by WB of cyclin E (marker of G1 and early S phase). This corresponds to the time point at which we have detected the highest expression of *IncREST* in the RNAseq data of synchronized HCT116 (**Fig. 2D**). At this stage, we didn't observe an increase of the DNA damage marker γ H2A.X, which reaches a peak in G2 in our experimental settings. The increased levels of γ H2A.X in G2 associated to cell cycle progression without DNA damage response has been already reported ((An et al., 2010); (Ichijima et al., 2005)). Therefore, we conclude that *IncREST* expression in HCT116 is cell cycle-dependent, regardless the presence of DNA damage in HCT116 cells. These data are now included in **Figure 2D**.

Figure R6. *LncREST* induction in synchronized cells is not due to DNA damage. **A.** Cell cycle of HCT116 at T=0 of the double thymidine block and at different release points; **B.** Flow cytometry profiles of HCT116 synchronized cells, representing the DNA content at different cell cycle stages from release (top) and relative expression of *IncREST*, *NDUFAF6* and *Cyclin E* measured by RNA-seq; log₂FC of each time point vs all other time points is represented. *p < 0.05 (mid); (bottom) Western blot of cell cycle markers Cyclin E, Cyclin A, and the DNA damage marker γH2AX normalized with total H2AX. **C, D, and E.** Quantification of the western blot in panel B for cyclin E (C), cyclin A (D) and γH2AX (E) protein levels.

- Fig 2A: please show the tracks from untreated and HU treated cells (total and chromatin for *IncREST*).

Now the RNA-seq tracks from chromatin and total samples (treated and untreated) have been added to **Figure 2A**.

Figure R6. *IncREST* gene locus annotation, chromatin acetylation and RNA expression. The RNA-seq is represented in counts per million (CPM)

- Supp Fig 2A: is *IncREST* chromatin bound also in untreated cells? Did the authors compared

the expression of *IncREST* by cell fractionation in untreated and HU treated cells? Please clarify the Supp Fig 2A.

IncREST is a chromatin associated RNA in HCT116 regardless replication stress challenges. The graph shown in **Supp Figure 2A** shows the cell fraction distribution of *IncREST* and other RNAs in non-treated HCT116. To clarify this concept, we have now added to **Supp Fig 2A** both NT and HU-treated condition. We also analyzed the cell cycle distribution of *IncREST* in APH treated cells (40 mM 24h) (as shown in the graph below), confirming the chromatin association of *IncREST* also in this condition.

Figure R7. *IncREST* is associated to the chromatin . Relative presence of the indicated RNAs in the different subcellular fractions.

- The authors should be consistent in representing the data in the main and supplementary figures. Please include individual values in all the graphs throughout the manuscript.

We have now added the individual values for the replicates in most the graphs (some did not allow this type of representation since the data are presented as relative distributions, such as cell cycle or subcellular localizations)

- Since RPE1 cells, a normal diploid cells, are widely used to study cell cycle and have a higher levels of *IncREST* (based on Supp Fig 2G) it would be worth of testing test whether treatment of RPE1 cells with HU and APH leads to upregulation of *IncREST* also in RPE1 cells. These two drugs are commonly used to induce RS in RPE1 cells.

We thank the reviewer for this suggestion. We have performed HU and APH treatment in RPE1 cells, as well as U2OS, as a further control, given that from the analysis of RNA-seq from the studies ((Yildirim et al., 2020), (Hao et al., 2020)) (in response to a previous comment from this reviewer) we observed a similar cell cycle dependent pattern of *IncREST* expression in both cell lines. As it shows in the graphs below, *IncREST* expression is induced by both treatments in both cell lines. We have added it now to **Supplementary Fig. 2**.

Figure R8. *IncREST* is induced by replication stress in different cell types.

- Fig 2D: the authors should show that no DNA damage is caused by double thymidine block to exclude that upregulation of IncREST in early S is not due to DNA damage, but rather its expression is cell cycle stage specific. The authors could also include qPCR for S phase genes such as cyclin E1 or D1 that are expressed in early S phase. The authors show provide H2AX staining for each time point to exclude DNA damage. This is an important experiment as the authors are showing that IncREST can be upregulated upon treatment of cells with different DNA damaging agents (Fig 2E).

We have now extended the analysis of the double thymidine block as suggested by the reviewer also in a previous comment, showing that no γ H2A.X increase is observed in early S phase, the time point corresponding to early S phase where *IncREST* is mostly induced in HCT116 cells. We have added western blot analysis of the markers of early, late, G2 phase to confirm the synchronization in Figure 2D, as extensively discussed in the previous comment of the same reviewer. Moreover, we added the expression of Cyclin E ((Mazumder et al., 2004)) as detected in the RNA-seq. The data are presented in the **Figure R6** and added to the revised **Figure 2**.

- Fig 3: Comet assay and micronuclei are not usually considered markers of RS, but rather markers of DNA damage. To show RS, the authors should perform IF for above mentioned markers.

We thank the reviewer for this observation. The data we are presenting shows that i) *IncREST* is induced by RS (inflicted with HU, APHI and CPT, **Figure 2E**) ii) is required for the correct activation of RS signaling (i.e. RPA recruitment and pATR, **Figure 3**), and fork stalling (**Figure 4**) iii) the consequence is increased DNA damage and mitotic defects (shown by comet assay, micronuclei formation and loss chromosomal congression, **Figures 3 and 7**), as consequence of an un-sigaled and un-repaired replication stress. In our view these results support the main conclusions of our manuscript, and we don't envision a plausible alternative explanation to unpaired RS signaling that is consistent with the data. We respectfully believe that although the proposed analyses will help characterize in more detail the phenotype of *IncREST* depletion, and in our view won't significantly add to the core findings of our manuscript.

- Fig. 3C: is the HU treatment ON? Why this treatment is different from the initial treatment (8hr)? Please include also untreated control cells upon IncREST depletion. The same is valid for H2AX foci in Fig 3D.

As explained in a previous comment, we have used different concentrations and times of treatment depending on the goal of the experiment. While we applied shorter treatment to have a better resolution of genes specifically induced by RS, and not by secondary pathways, longer times are required to better appreciate the accumulation of signals of damage, such as γ H2AX. Knowing that *IncREST* is consistently activated by HU in a time and dose dependent manner, we have then used harsher conditions to better appreciate certain phenotypes, but always according to what is generally performed in literature for each type of experiment (see also previous responses to this reviewer). However, since this is a concern for this reviewer, we have repeated the micronuclei analysis by treating cells with 1mM HU for 8h instead of overnight. We have now added also the NT conditions and have updated **Figure 3C** (see **Figure R9**)

Figure R9. The knockdown of *IncREST* leads to an increase in the number of micronuclei. micronuclei in *IncREST* KD HCT116 cells in NT and HU treated condition (1mM 8h). Top panel, representative images with zoom-in on one cell bearing micronuclei per sample. Scale bar: 20um. Lower panel, percentage of cells with one or more micronuclei. * $p < 0.05$.

Moreover, we have now added complete data of γ H2A.X foci analysis presented in in **Figure 3** showing both NT and HU treated condition. The complete panel has been added to the Figure, with relative foci analysis of triplicates.

Figure R10. The depletion of *IncREST* leads to a decrease in gH2AX levels in conditions of replication stress. Representative images and quantification (dotplot and median) of gH2AX foci per nucleus in HCT116 cells transfected with CTRL or *IncREST* LNAs, NT and HU (1mM o.n). Scale bar, 10 μ m. At least one hundred cells were analyzed per sample in two independent replicates. * $p < 0.05$, ** $p < 0.01$, **** $p < 0.001$ (Mann Whitney Wilcoxon U-test).

• Fig 3C: The micronuclei look slightly bigger that they should be. Please zoom into one cell for each condition. Micronuclei are usually around 2-2.5 μ m in size, but these nuclear structures look bigger. Is it possible that some of the micronuclei scored in this assay might be nuclear blebbing caused by apoptosis? Can the authors stain micronuclei with lamin B1 and H2AX. I am

expecting that these micronuclei will have no lamin B1 staining and high levels of DNA damage. Please show the graph also for untreated cells upon IncREST depletion to have an idea of basal level of micronuclei formation without HU. I suggest to show untreated and HU treated cells for every IF image, as done with CRISPR-mediated depletion and H2AX staining in Supp Fig 3. Please specify how many experiments were performed for that staining (Supp Fig 3C-D).

It is our understanding that the size of micronuclei is not a very well-defined parameter, and that they can be of various sizes. Even if the average size is 2-2.5 μm as the reviewer pinpoints, they can range from 0.8 to 4 μm ((Chmielewska et al., 2018)). Moreover, micronuclei size is proportional to the originating nucleus, which can change based on the cell type. In this regard, micronuclei range from 1/5 to 1/20 of the original nucleus size ((Ye et al., 2019), doi.org/10.1016/bs.mie.2019.05.015, doi.org/10.1016/B978-0-12-409547-2.12381-9). We considered the concern of the reviewer that some of the micronuclei scored in our analysis looked slightly bigger than what a micronucleus should be, therefore we have repeated this experiment, using the HU condition suggested by the reviewer in a previous comment. We scored only those micronuclei that showed a maximum diameter of 3 μm , avoiding the bigger particles that are of uncertain attribution. We added this panel to **Fig. 3C**. As suggested by the reviewer, for each condition we have zoomed in one cell per image to better show the ratio between the nucleus and the relative micronucleus (See **Figure R9**, and previous comment).

Below is an example of how we scored for micronuclei based on their size.

Figure R11. Lamin B1 is a known component of micronuclei, and its absence or depletion is generally related to micronuclei disruption. We think that assaying for the presence/absence of lamin B1 in micronuclei, although interesting, is out of the scope of this paper. The presented data supports that *IncREST* depletion, especially during the RS challenge, alters the genetic stability of HCT116 cells, increasing the formation of these structures that are generally assayed in immunofluorescence by DAPI staining.

We have added NT condition in Figure 3D, (IF of $\gamma\text{H2A.X}$) as requested also in a previous comment by this reviewer (see previous paragraphs).

- Do CRISPRi and CRISPR KO cells for IncREST have higher rate of micronuclei upon HU? The

same question is for defects in ATR/CHK1 signalling. In addition to IF, defects in ATR/CHK1 signalling are usually shown by western blot. Thus, the authors should confirm defects in ATR/CHK1 signalling by using antibodies for total ATR, ATR phospho, CHK1 total, CHK1 phospho and RPA phospho pRPA 32 (especially if these cells have a ssDNA gaps) upon HU (and APhiI) after LNA, CRISPRi and/or CRISPR KO cells for *IncREST*. Alternatively DNA fiber assay should be performed with one of the CRISPR methods.

We have verified the defects in the stress signaling by immunofluorescence of gH2A.X in the CRISPR KO and CRISPRi cells (**Supplementary Fig. 3**), indicating that that under conditions of replication stress, the signaling is affected. We also show that *IncREST* KO cells present increased levels of DNA damage as measured by comet assay when treated with HU, but not in untreated conditions. (**Supplementary Figure 3J** and **Figure R12**). Moreover, following the reviewer request, the defective activation of the S-phase checkpoint in *IncREST* KO cells is shown by DNA fiber assay. *IncREST* CRISPR knockout cells show extended DNA fibers and faster fork speed in conditions of HU (**Figure 4E**). Importantly, the ectopically expressed *IncREST* is able to recover the fork speed in *IncREST* KO cells at the same level of WT cells (**Figure 4E** and **Figure R12**).

Altogether, these results strongly support that the lack of normal *IncREST* expression is linked to an impaired fork signaling upon RS.

Figure R12 Comet assay and quantification (top) and fiber assay on *IncREST* WT and KO cells (bottom).

- Fig 4A: the authors should clarify why the 50uM Hu was used, and why HU was added during the second pulse and not after Cldu?

Referring to a previous comment of this reviewer, we take the opportunity to clarify the experimental approach we applied in the replication fork analysis that we show in Figure 4. As we have previously commented, micromolar doses of hydroxyurea have the capacity to strongly impair fork progression without causing S-phase arrest (PMC3896206, 10.1038/s41464-023-

37341-y; 10.1038/s41588-022-01023-0). This is essential to appreciate the differences in the replication potential of cells through the technique of the fiber assay. A similar experimental setup is commonly used by several groups that are well known experts in replication studies (e.g. 10.1038/s41464-023-37341-y; (Nakatani et al., 2022)). Specifically, we have used 50 μ M HU for the fiber assays as low concentrations of HU allow to detect subtle alterations in the replication progression. The fiber assay experiment would not otherwise allow us to detect small alterations of the replication pattern, as in the short times that are used for the experiment (20 minutes pulses), high concentrations of the drug would block the replication forks to the point that is beyond the detection ability of this assay. Indeed, our intention with this experiment was not stalling the replication fork, but rather slow down the replication fork speed to study the behavior of *IncREST* depleted cells vs control cells. Specifically, we meant to add HU in the second pulse to detect the different behavior under induced RS for two reasons: 1) to catch the behavioral switch of *IncREST* depleted cells before/after HU treatment, and 2) even if we see an effect of *IncREST* depletion in NT cells, as shown by several experiments, the strongest alterations are observed in presence of an exogenous RS source, like HU treatment.

- Fig 4C: the authors should demonstrate the same graph but without S1 endonuclease; that is a standard for this experiment.

As requested by the reviewer, have revised **Figure 4** by adding the control experiment performed under the same conditions but without S1 adding endocnuclease in **Figure 4C** (and **Figure R4**). In the absence of S1 nuclease, the IdU tracts are longer in *IncREST* KD cells, confirming what we have observed in the experiment using 20' pulses. However, when we add S1 nuclease, the length of the second pulse is reversed, confirming a discontinuous replication in *IncREST* KD cells compared to control cells.

Figure R13. *IncREST* depletion bypasses fork arrest with discontinuous replication. New panel C of Figure 4. Experimental setup for replication fork analysis of control or *IncREST* KD cells \pm S1 nuclease, and dot plot and median of CldU/IdU ratio. *** $p < 0.001$, **** $p < 0.0001$, by Mann-Whitney test.

- Fig 5: Why 3mM HU was used for iPOND assay. It is not clear why different HU concentration are used throughout the manuscript.

We apologize for the lack of information explaining the rationale behind the experimental conditions used. The use of different HU treatments in different experimental settings is motivated by the type of experiment and the protocols previously established by other labs. In the specific case of iPOND experiments, we used 3mM HU to induce fork stalling, following a procedure established by the group that first set up this technique (Sirbu et al., 2013). In contrast to the fiber assay, where we did not want to completely stall replication, ensuring a complete block of replication for a given time is of utmost importance for iPOND, to allow an accumulation of protein and RNA players of the replication fork repair on the stalled fork. Moreover, as described in detail in the methods part related to this experiment, there are technical challenges that need to be considered: cells have to undergo several pulses, therefore it would be unviable to perform a HU pulse of 1mM for 8h.

Another specifics of the iPOND experiments, compared to others performed in this paper, is cellular confluence. We generally perform our experiments by having cells at a starting 60-70 % confluence (e.g. to have a minimum cell overlap in IF experiments and not to have stressed confluent cells in general). However, in order to obtain the sufficient number of cells for iPOND and the right conditions, cells are seeded some days before the experiment, synchronized and released, and at the moment of the pulses they are usually more confluent than the cells treated with 1mM HU as in Figure 1. As it is well known, any treatment needs to be optimized based on the number and confluence of cells, and in this case, we used HU conditions that give a similar induction of *IncREST* given confluence and time available for the experiment.

- The authors should show quantitative comparison of HU-treated cells in control (luc) and *IncREST* pulldown (either by SILAC or TMT). Showing the graph with peptide intensities is not the standard in the proteomics as it is possible that NCL is also present in the control (luc) cells. Please show the data as volcano in HU treated cells to indicate significant interactors.

For the design and interpretation of the masspec experiments, we have followed the guidance of the experts of the Taplin Facility at Harvard, where the masspec analysis was performed. They recommended to use a label free method, since SILAC and TMT tend not to give the same level of sensitivity for the detection of proteins, as reported in the literature (Chen et al., 2021), and since in *IncREST* pulldowns we are only able to retrieve low amount of proteins. Moreover, label-free methods offer other advantages, as they provide more efficient protein identification and quantification, and a higher dynamic range of quantification than the stable isotopic labeling approaches.

In addition to these technical considerations, it should be considered that masspec was used as the initial approach for protein identification, but most importantly, the interaction between *IncREST* and NCL was independently validated by RNA immunoprecipitation (RIP) and RNA in vitro pulldown (Figure 6A-B). Moreover, we were able to validate additional *IncREST* interactors detected by the masspec, such as PCNA, SSRP1 and TRIM28 (Fig R14).

The full list of proteins and the relative numbers of peptides identified for each sample is provided in Supplementary Fig. 5 and Supplementary Data 2. Since we are not using a quantitative but

qualitative method for the masspec analysis, we cannot define *p*-values and therefore the volcano representation is not possible.

Figure R14. Independent validations of masspec analyses. RIP experiments with TRIM28, SSRP1, PCNA and IgG control antibodies, detecting the enrichment of *IncREST* or *HPRT* as control.

- The authors should show NCL enrichment in Fig 5B by western. Is NCL recruited to chromatin upon HU where it could interact with *IncREST*?

We now show the enrichment of NCL on chromatin upon HU treatment in **Figure 6C**

Figure R15. *IncREST* increases its chromatin localization upon replication stress treatment

- Fig 6D: What is the pattern of NCL staining in untreated cell (no *IncREST* depletion). It is important to show NCL staining in untreated and treated cells (HU).

We have now added the non-treated condition for the immunofluorescence of NCL in *IncREST* depleted cells. We see that NCL is more nuclear upon treatment, which is reduced upon *IncREST*

KD. Importantly, global levels of NCL don't change across these conditions (Fig 6E and Supplementary Fig.6A-B and Figure R16).

Figure R16. *IncREST* controls NCL nuclear sub-localization. IF images of NCL detection in control or *IncREST*-depleted cells (left). Quantification of the ratio of NCL staining between the nucleoplasm and nucleoli. In the bottom of the figure, the total levels of NCL protein is shown by western blot.

- Fig 6E: is this untreated or HU treated cells? Please indicate in the Figure. What is the level of RPA32 in untreated cells? From Fig 6G it is clear that *IncREST* depleted cells without any damage have less RPA in the input. These data suggest that *IncREST* depleted cells are experiencing defects in replication even in the absence of any RS.

According to this Reviewer's request and also from Reviewer 2 we have added the WB and its quantification for the NT samples. The WB that the changes in the chromatin association of these proteins mainly occurs under conditions of treatment.

Figure R17. Western blot image (left) and quantification (right) of NCL and RPA32 in the chromatin fraction of *IncREST* depleted cells and control LNA in NT and HU treated condition

- Fig 7: What is the cell cycle profile of IncREST depleted cells? It is not surprising to find less interaction between RPA and NCL upon HU since IncREST depleted cells have lower RPA levels in untreated cells, and are proliferating less.

We have performed cell cycle analysis of *IncREST* depleted cells in NT and HU conditions. We did not observe a significant change in cell cycle of non-treated cells, however, when we deplete *IncREST* and induce replication stress with HU or APH, we observe some increase of cell population in G2 phase (see **Figure R1**). However, we don't see global expression changes of RPA nor NCL (**Supplementary Fig. 6A-C** and **Figure R18**) when *IncREST* is depleted, which supports the idea that the differences observed in their interaction is not due to changes in protein abundance, but to the localization changes observed.

Figure R18. NCL and RPA32 decrease their chromatin localization upon *IncREST* depletion in HU-treated cells. A. NCL and B. RPA32 proteins detected by WB and quantified in total extracts or chromatin fractions of HCT116 treated as indicated.

- Is RS occurring in nucleoli upon HU in IncREST depleted cells? Do authors find TOPBP1 and Tracle in their proteomics experiment (see PMID: 34100862)?

We cannot exclude that replication stress is also occurring at nuclei. However we don't have data to support it either. *IncREST* FISH pattern is not compatible with a nucleolar localization of the lncRNA (**Figure 2C**), and we did not identify TOPBP1 or TRACLE in our proteomic analysis presented in **Supplementary Fig. 5** and **Supplementary Data 2** where we are displaying the full list of proteins identified in the proteomics experiment. We have performed IF experiment of TOPBP1 in *IncREST* depleted cells and relative control, NT and HU treated condition, but we were not able to detect a pattern of nucleolus staining for this protein.

- If IncREST is involved in RS response, that IncREST depleted cells should be more sensitive to ATR, Wee1 or CHK1 inhibitors. Did the authors tested the sensitivity of these cells?

We thank the reviewer for this suggestion. We have performed cell proliferation analyses (MTS) on *IncREST* depleted cells treated with AZD6738, a specific inhibitor of ATR at different concentrations for 72h. Even at lower concentrations of ATRi there is a reduced sensitivity of

CTRL LNA cells to the drug. At higher concentrations the differences are reduced, probably due to the high toxicity of the drug, which suggest that the inhibition of *IncREST* could be beneficial as combined therapy using low doses of ATR inhibitors. Since these are preliminary results that we would like to further develop in a future study, we haven't included them in the revised manuscript.

Figure R19. Cells depleted of *IncREST* increase are more sensitive to ATR inhibitor. MTS assay of *IncREST* depleted cells with LNA GapmeRs and relative control treated with ATR inhibitor AZD6738. Cells were seeded in 96 well plate and transfected. 48h post-transfection they were treated with different concentrations of ATRi and incubated for three days before the MTS assay was performed

- Fig 7A; to show the link to CIN, the authors need to demonstrate that *IncREST* depleted cells have anaphase cells with lagging chromosomes or chromatin bridges. Please check the figure legends, as the text is not corresponding to the images.

Please refer to previous responses to a similar comment.

- Fig 7B: The quality of images showing mitotic defects needs to be improved. How did the authors define misaligned chromosomes? It is not clear from images in Fig 7 that CENPA signal is outside of the metaphase plate upon *IncREST* depletion. In addition, the cells seems to have problems with microtubules (increased microtubule stability was shown to be associated with RS). It would be interesting to show whether these cells have anaphase cells with lagging chromosomes or chromatin bridges (it is possible that the cells will arrest in metaphase due to spindle defects). Scale bar is missing.

We agree with the reviewer that chromosome congression is a phenotype that is relatively difficult to quantify. Considering the schematic representation of the mitotic cell with chromosomes aligned on the spindle pole (draw a schematic like in the figure below in the middle), and based on previous reports (Ma et al., 2007) and as we also describe in the methods section, we counted the chromosomes that are located outside the tubulin spindle. Specifically, we counted the CENPA signals that were located outside the tubulin pole for each mitotic cell analyzed.

- What is the difference in mitotic defects shown in Supp Fig 7B and in Fig 7B?

Fig. 7B is showing mitotic defects in *IncREST* depleted cells, while Fig. S7B is showing the same type of defect for NCL depleted cells. The latter is a known phenotype arising in cancer cells when NCL is depleted, and is considered a cause of apoptosis and mitotic catastrophe. We will adjust the text to make sure it is clear in the new version of the manuscript.

- Based on Drows et al, RS is associated with eight signatures suggesting it is a major source of CIN. In addition to CX9 signature, there are other signatures (eg CX13, CX8, CX11) (line 315) that did not correlate with low RS. What could be the reason?

The aim of our analysis was to determine whether there is a connection between *IncREST* expression and genomic instability in cancer. Drows et al. defined the CIN signatures based on computed distributions of five fundamental copy number features that were previously demonstrated to represent different underlying causes of CIN. Then data was integrated (including gene expression and mutation of cancer driver genes among others) to propose the putative aetologies underlying each signature. This assessment was done with different levels of confidence, as expected, given that the signatures are defined with data from thousands of tumors across multiple cancer types, naturally involving a high heterogeneity. It is therefore difficult to speculate why other replication stress signatures showed an association with *IncREST* expression changes, and even more difficult to derive mechanistic conclusions from such analyses.

CX8, CX9, CX11 and CX13 encoded patterns of low-level, mid-level, mid-level and high-level amplifications, respectively. Higher activity of CX8 in the context of amplification and overexpression of U2AF1 and MAPK1, and for CX9 ERBB3, suggested replication stress as a putative cause. All four signatures were associated with increased cell cycle score, reinforcing replication stress as a causal factor. In addition, CX8, CX9 and CX13 were associated with APOBEC mutagenesis. CX9 copy number changes were not part of oscillating chains; however, the remaining amplification signatures were. CX13 was strongly associated with extrachromosomal DNA circularization and amplification events; however, the specific mechanism causing the extrachromosomal DNA was not evident. CX9 was correlated with response to multiple kinase inhibitors targeting genes involved in major mitogenic pathways (EGFR, JAK1, MET, PRKCA and PIK3CA), suggesting that a multikinase inhibitor approach may be suitable for targeting this group of tumors.

- Sup Fig 7B- scale bar is missing. It seems to me these cells have multipolar spindles based on tubulin staining, is that the case?

Perhaps the reviewer is referring to the spindle of LNA1 KD? We believe it's just slightly on a different plane than the other two shown, which are more clear. But overall in the analysis we did not appreciate a mitotic spindle defect.

The scale bar was added to the figure.

Minor corrections:

We appreciate the reviewer for detecting these errors that we have now corrected.

- Fig 6F: there is a mistake in the image, the staining is for NCL but quantification is showing RPA32. Please correct.

This has been replaced

- What is Fig 6B showing? Please indicate the western blot in the image.

We have revised Figure 6

- Include catalogue number of the drugs and reagents used in this study (line 425).

This has been added.

- FOXM1 is a transcription factor with a known role to regulate expression of genes in G2/M, therefore I would tune the sentence (line111) that FOXM1 has a well-known role in RS (based on one paper cited). This should be corrected in the text.

Indeed, FOXM1 is known to control the expression of genes involved in G1/S transition and replication, but also G2/M and mitosis (<https://doi.org/10.1016/B978-0-12-407173-5.00004-2>). We will rephrase it accordingly

- Did the authors try smRNA fish using Stellaris FISH probes as they did in the past ?

In this study we used FAM-labeled Locked Nucleic Acid (LNA) DNA probes.

- Line 551: should be HU not "hu".

ok

- Line 726: should be RNA-seq not RNAseq

ok

- Include page numbers in the manuscript.

ok

Reviewer #2 (Remarks to the Author):

Statello et al. describes the involvement of IncREST in the cellular response to replication stress. The paper is well-written, the data and methods are comprehensive, and the conclusions are, for the most part, supported by the presented data. The identification of IncREST and its association with NCL is very convincing, as well as the impact of IncREST for genome stability and tumor growth. Mechanistically, the function of IncREST is more difficult to assess, as the involvement of its binding partner NCL itself in replication stress is not established. Nevertheless, the results are novel, interesting and important. I feel the manuscript is appropriate for the broad readership of Nature Communications but there are a few major issues that I believe need to be clarified before publication.

We thank the reviewer for considering our work novel, interesting and important.

Here are my detailed comments:

Figure 1

1A: Add total protein for ATR, p53 and H2AX in WB

We have performed these WBs and we will add the total counterparts of pATR, p-p53 and γ H2AX to the WB panel in **Supplementary Fig.1A**

1B-C: Ok

1D: Is the figures showing total or chromatin-associated RNA, or both?

The figure is showing chromatin associated RNAs

1E-F: Ok

1G: Include color explanation in figure or legend.

We added the explanation in the legend

Figure S1A-C: Ok

Figure 2

Figure 2 has been revised

2A: Ok

2B: Typo = citoplasm should be spelled cytoplasm

2C: Are these dots of IncREST corresponding to sites of replication stress? I would be interesting to combine the FISH staining +/- HU with staining of RPA and NCL

We tried to perform this experiment as suggested by the reviewer. FISH-immunofluorescence is technically challenging even more for IncRNAs, which are typically lowly expressed. It is difficult to find probe hybridization conditions compatible with the use of antibodies. Despite multiple attempts we failed to obtain simultaneous signal for *IncREST* and RPA or NCL. However, although we could not visualize it, we have shown that *IncREST* is enriched at sites of nascent DNA upon replication stress (iROND experiments) and under these conditions it is associated with factors implicated in the response to replication stress (Masspec analyses). The iROND experiments are done under astringent conditions, where cells are synchronized,

and only the nascent DNA is labeled. Moreover, controls +/- treatment and nascent vs mature chromatin are shown. On the other hand, masspec experiments are performed by pulling down the endogenous *IncREST* after crosslinking in treated vs untreated cells. In our view these data are more robust than images obtained by low resolution microscopy. Even if we could provide these images, they wouldn't be able to distinguish whether the interaction occurs at the sites of stress, or in their proximity, for which we would need specialized high resolution microscopy, which unfortunately is not available to us.

2D-G: Ok

Figures S2A-H: OK

Figure 3

3A: Ok

3B-C: Show comet assay and micronuclei number also in no-HU conditions.

We have performed comet assay in NT, HU and APH condition and present this data in **Figure 3C (Figure R20)**, and micronuclei analyses (**Figure 3D and Fig R21**). The results consistently show increased damaged linked to *IncREST* depletion.

Figure R20. Comet assay in HCT116 cells treated with the indicated drugs an LNAs targeting *IncREST* or control.

Figure R21. New panel C of Figure 3: micronuclei in *IncREST* KD HCT116 cells in NT and HU treated condition (1mM 8h). Top panel, representative images with zoom-in on one cell bearing micronuclei per sample. Scale bar: 20um. Lower panel, percentage of cells with one or more micronuclei. * p<0.05.

3D: The reduction of gH2AX is strange and cannot solely be explained by an attenuated activation of ATR. Is ATM also impaired following loss of IncREST and/or siNCL?

This an interesting question that we tried to address. Unfortunately the WBs for ATM did not work in our hands.

3E: It would be good to assess levels of gH2AX, pATR and pATM (as well as total counterparts) after LNA IncREST employing western blot (WB). Include both NT and HU conditions. Assessment of protein levels by WB and also in no-HU conditions is important through the manuscript.

We have performed WB for the replication stress markers in *IncREST* depleted cells in NT and HU treated condition. The results confirm that, as shown by IF, the pATR, pCHK1 and gH2A.X are decreased when the cells are treated with HU. Unfortunately in our hands the WB for ATM did not work. This has been added to Supplementary Fig.

Figure R23. WB on different replication stress markers in *IncREST*-depleted cells.

Figures S3A-B: Ok

Figures S3C: Are the stable KO/CRISPRi cells also showing signs of DNA breaks and/or micronuclei?

We show now that CRISPR KO cells also present increased DNA breaks in **Supplementary Figure 3J (Figure R23)**

Figure R23. Comet assay images and quantification in control of *IncREST* CRISPR KO clones under the indicated treatments.

Figure S3D-J:Ok

Figure 4

Is loss of *IncREST* affecting cell cycle distribution?

IncREST depletion causes an accumulation of cells in G2 phase after treatment with both HU and APH, which is consistent with the abnormal progression through the S-phase, since *IncREST* KD enables replication of damaged DNA and elicits progression of cells through the S phase, leading to G2 arrest resulting from persisting DNA damage. However, we did not find significant cell cycle alterations in non-treated *IncREST* depleted cells. We will add these data to the manuscript as suggested by this reviewer and reviewer 1

Figure R24. Cell cycle profile of HCT116 cells untreated or upon RS treatments (1mM HU for 8hrs, 40uM APH 24h) and transfected with the indicated LNAs.

Figure 5

5A: Ok

5B: Is NCL among the proteins obtained by iPOND? The blot for RAD51 is dirty and bands cannot be assessed properly

NCL is detected in the iPOND, although shows its presence in all the conditions, (nascent and mature chromatin). We have repeated this several times with similar results, observing sometimes even presence of NCL in the no-click condition (**Figure R25A**). We suspect that the buffer used for the iPOND experiment (with high concentration of Cu(II), 2mM) favors aggregation and precipitation of NCL, which is a protein with two IDRs and tendency to aggregate. This effect of Cu(II) has been indeed reported in the literature for other disordered proteins (John et al., 2021). We believe that this is the reason why this experiment is inconclusive. On the other hand, we can observe the enrichment on NCL in the chromatin upon HU treatment (**Fig R25B**), which we now include in the revised version of **Figure 6**. We conclude that NCL is enriched on the chromatin, but not exclusively at the replicating forks.

We are also including a better WB image of RAD51 in the iPOND in **Figure 5**.

Figure R25. A. Image of WB of inputs (top) and experimental samples (bottom) from iPOND experiments. **B.** WB image (top) and quantifications (bottom) after cellular fractionation in the indicated conditions. N= 3 biological replicates. **p < 0.01, ***p < 0.001, ****p<0.0001 (mean ± SD, two-tailed unpaired t-test).

5C-F: Ok

Figures S5A-D: Ok

Figure 6

More controls linking NCL and replication stress would strengthen the study.

We have now included new data showing the change of NCL localization upon HU treatment, as well as further phenotypical analysis showing that the depletion of NCL recapitulates the signaling phenotypes: i.e defective recruitment of RPA, pATR and gH2AX, linked to increased DNA damage measure by comet assay. All these data are shown in Figure 6 and Supplementary Fig. 6

6A: Ok

6B: The NCL label in the top blot is missing.

The Figure has been updated

6C: To assess how much of the chromatin-bound NCL that reflects binding to stalled replication forks, a WB should be included showing NCL binding to chromatin +/- HU. Also include LNA IncREST in both +/- HU in this blot.

As suggested by the reviewer, we have performed western blot analysis of NCL associated to cellular fractions in HCT116 cells +/- HU. NCL association to chromatin increases following replication stress, while total NCL is not affected this is the quantification of the experiments. These results are now included in the revised **Figure 3C** and **Figure R25**.

Moreover, we have performed WB of NCL in chromatin fraction of *IncREST* CTRL cells and *IncREST* KD before and after HU treatment, in the same style as RPA is shown in **Supplementary Fig 6A**. Following the reviewer's suggestion that this representation is more informative than what we were previously showing in Figure 6C, we have added this panel in **Fig. 6D**.

Figure R26. WB image on chromatin of NCL and H3 as loading control in the indicated conditions. The quantifications is represented in the bottom N=2.

6D: It would be good to include evidence that NCL changes localization upon replication stress (i.e., +/- HU), to more firmly establish that *IncREST* facilitates this movement. Now only +HU is shown

We thank the reviewer for this suggestion, besides the experiments just described, we have added the NT condition to the immunofluorescence shown in **Figure 6E** and **Figure R27** It is important to emphasize that the total levels of NCL don't change upon replication stress or *IncREST* knockdown (WB) (**Supplementary Fig 6I** and **figure R28**)

Figure R27. IF images showing the nuclear localization of NCL (left) and quantification of the relative ratio of the nucleoplasmic and nucleolar staining (right)

Figure R28- Total NCL protein levels.

- At text row 275, the authors write “p-ATR, recruitment that we have found to be affected”. This has not been shown, only reduced levels of pATR. Either examine recruitment using IP or IF of pATR followed by co-precipitation or co-staining of RPA, or re-phrase this sentence. We have rephrased this statement

6E: It would be good to include chromatin-association of RPA32 in IncREST-depleted cells also in no-HU condition (similar blot as S6A).

We have now included this experiment in **Figure 6D** (see also **Figure R26**)

What is the reason the reduced levels of RPA in chromatin following knockdown of IncREST? Impaired generation of ssDNA or reduced binding of RPA to ssDNA? This is important to explore, as the reduced levels of RPA in chromatin likely underlies the attenuated pATR signal.

The fiber assays combined with nuclease S1 (specific for ssDNA) demonstrate that *IncREST*-deficient cells have increased ssDNA breaks, since the S1 cuts the fibers of *IncREST*-depleted cells. This is corroborated by the comet assays, also showing increased DNA breaks. This is consistent with the notion that the decrease of RPA binding is not a consequence of a decrease of ssDNA, but rather its impaired recruitment to the sites of damage.

6F: I think the NCL legend on top of middle image should be replaced for RPA32

Thanks for noticing this. The figure has been revised.

6G: Ok

S6A: This WB is very informative as it includes both NT and HU conditions.

S6B: Ok

S6C-D: Show a WB of chromatin-associated RPA32 after siNCL +/- HU and in a similar style as the blot in S6A. Also blot for pATR and gH2AX, and total counterparts.

We now show the requested WBs in **Supplementary Fig. 6** and **Figure R29**

Figure R29. Changes in RS markers upon NCL knockdown.

S6E-F: Ok

Figure 7

There seem to be a mixup with main, suppl figures and legends here.

Thanks for noticing the mistake. It has been corrected now.

Is expression of IncREST showing any correlation to patient survival? If downregulation of IncREST kills cancer cells one would think that tumor cells would avoid this, or?

As suggested, we have performed the survival analysis using the publicly available TCGA datasets, although we have not observed significant correlations with IncREST expression.

Figure R30. Survival of TCGA colorectal cancer (COAD) patients with high and low *IncREST* expression levels.

A model figure would be good to include.

We include now a model in **Figure 7**.

Figure R31. Model proposed for *IncREST* function. Upon replication stress, NCL protein localizes to the chromatin where it interacts with the induced *IncREST*. This is required for the localization of RPA and pATR at the forks for the effective S-checkpoint signaling linked to fork stalling. The action of *IncREST* protects cells from genomic instability by assuring the correct functioning of the S-phase checkpoint.

The authors propose that loss of *IncREST* results in replication stress. However, the signs of replication stress, i.e., elevated RPA in chromatin, elevated pATR and gH2AX are not seen after LNA *IncREST*, but instead a reduction of these factors. What could be the mechanistic explanation of this? This should be discussed in the manuscript.

We apologize for not stating clearly our model. Our data show that the loss of *IncREST* impairs the proper signaling upon replication stress, which normally would result in elevated RPA, pATR and gH2AX. Therefore, *IncREST* is required for the correct response and genomic integrity.

Although we don't fully understand the mechanism behind, based on our data we propose that *IncREST* helps the nucleation of factors that signal at the damaged fork. We speculate that the formation of molecular condensates will favor the efficient recruitment of signaling factors. *IncREST* may be one of the components of such condensates involving interactions with NCL (NCL contains two IDRs that confer it with the capacity to form biomolecular

condensates) and possibly other proteins. This is further corroborated by the ability of *IncREST* to co-purify together with components of the replication stress response other than NCL, which are known to interact together in the establishment of the replication stress response (Figure 5 and Fig. R14). We envision that besides *IncREST*, other ncRNAs could be part of such subnuclear phases, helping orchestrate the replication stress response.

Reviewer #3 (Remarks to the Author):

In this manuscript, Statello et al seek to identify lncRNAs that are induced by replication stress, accumulate in chromatin, and might participate in the cellular response to stalled forks. They focus on *IncREST*, which harbors regulatory motifs for transcription factors involved in the replication stress response, including a p53 response element. The authors use LNA, CRISPR gene deletion KO and CRISPRi as three alternative LOF approaches and observe mitotic defects, increased apoptosis, and, as a result, impaired proliferation. Overall, this manuscript identifies a new lncRNA (see point 1 below) and identifies cellular phenotypes associated with perturbation of various functional elements in its locus. However, revisions are requested to address the potential cis regulatory function of the lncRNA locus (point 2) and the potential functional but non-specific interactions with protein factors (point 3).

1) The authors refer to this lncRNA as a novel and uncharacterized lncRNA, giving it the name *IncREST*. However, the same lncRNA was recently described and named p53-regulated tumor-suppressive lncRNA) by another study (Mitra, Adams, and Eischen, Cancer Res Comm 2023). The authors should update their manuscript accordingly to reflect the published name.

We thank the reviewer for bringing this paper to our attention, which we had overlooked. We have now included a reference to it in the revised version of the paper.

2) It is very strange that the authors provide expression data for *NDUFA6* but not *TP53INP1* RNA levels in Fig. 2, especially given that the p53 peak is shared between the two genes and given the effects of LNA and CRISPR KO/CRISPRi on *TP53INP1* levels, discussed in Supplementary Figures.

We apologize for this oversight. We now include the expression of *TP53INP1* in Figure 2A. In addition to that, we show the expression of *TP53INP1* in different datasets of cell-cycle synchronized cells, as well as in cells treated with different drugs. All these data are presented in the revised Supplementary Fig. 2D-F, H. As expected, *TP53INP1* presents a similar expression pattern and regulation as *IncREST*.

The effects of *IncREST* LOF perturbations on *TP53INP1* are concerning and suggest a role for this lncRNA locus in regulating *TP53INP1* expression, which in turn may be the driver of the S-phase phenotypes. This should be addressed in a revisions. The opposite effects of *IncREST* LOF perturbations on *TP53INP1* may simply reflect the simultaneous presence of stimulatory and inhibitory elements in the locus that are perturbed differentially by the perturbations, as recently seen for other lncRNAs (ie lncRNA Meteor, Cell Reports 2023).

We agree that the opposite effects of *IncREST* KD on *TP53INP1* probably represent different local effects that are dependent on the type of perturbation applied (i.e. ASO, CRISPRi, or CRISPR KO). However, despite this fluctuations in *TP53INP1* expression, the replication stress phenotypes are consistent, always correlating with *IncREST* downregulation and independent of the direction of the changes in *TP53INP1* expression. Moreover, this protein coding gene is known as a regulator of autophagy (Seillier et al., 2012), and not known to be involved in replication stress.

Nevertheless, we have taken into consideration this important comment and further investigated the relationship between *IncREST*, *TP53INP1*, and the checkpoint phenotypes. When *IncREST* is depleted by LNA, the expression of *TP53INP1* is induced by approximately two fold (**Supplementary Fig. 3D**). To evaluate whether the checkpoint phenotype could be mediated by the induction of *TP53INP1*, we performed experiments by overexpressing *TP53INP1* from a plasmid, which unfortunately leads to the downregulation of *IncREST* (**Fig. R32A**) Under these experimental conditions we also observe an increase in the number of cells in G2 phase (**Fig. R32B**), in line with the previously observed for *IncREST* reduction (**Fig. R1**). Upon *TP53INP1* overexpression and CTP or HU treatments we also observed some reduction of the checkpoint marker p-ATR (**Fig. R32C**), similar to what we observed with *IncREST* KD. Therefore, since in both experimental conditions *IncREST* is downregulated, these experiments do not allow us to unequivocally ascribe the effects to the induction of *TP53INP1*. Interestingly, we also observed that *TP53INP1* protein is very lowly expressed, with undetectable endogenous levels (**Fig. R32C**), which argues against a significant role of the protein in the experimental conditions of our study. This can be even more relevant in the context of the experiments of *IncREST* KO and CRISPRi, where *TP53INP1* gene is further downregulated (**Supplementary Fig. 3D-F**), however the checkpoint phenotypes are the same (**Supplementary Fig 3J-L**) and consistent with the downregulation of *IncREST* in each condition.

To further investigate the possible mutual regulation of *IncREST* and *TP53INP1* genes, we performed the individual KD of *IncREST*, of *TP53INP1*, or the double KD. As previously observed, the KD of *IncREST* leads to upregulation of *TP53INP1*, however *TP53INP1* KD leads to downregulation of *IncREST* (**Fig. R32D**), and increased number of cells in G2 phase (**Fig. R32E**), again consistent with the observed in the other experimental settings when only *IncREST* is targeted. Although these experiments do not allow us to clarify the effect of *IncREST* on *TP53INP1* regulation, given the opposite direction of the changes in *TP53INP1* expression upon *IncREST* depletion by LNA (up) vs CRISPR KO and CRISPRi (down), it seems plausible that the checkpoint phenotype is linked to *IncREST* rather than *TP53INP1*.

More importantly, (i) *IncREST* enrichment at replication forks (**Fig 5C**), (ii) the capacity of the lncRNA to rescue the aberrant fork progression when expressed in trans (**Fig. 5E**), and (iii) the dependency on its interaction with the protein NCL (**Fig. 5F**), which is involved in the same pathway (**Fig. 6 and Supplementary Fig. 6**), support a direct role of *IncREST* in this response. However, we are aware that the scenario can be far more complex, and *IncREST* may have additional biological effects through the regulation of *TP53INP1* expression. We have expanded the discussion to include these considerations.

Figure R32. *IncREST* and *TP53INP1* reciprocally modulate one another's expression. **A.** *IncREST* expression levels in cells transfected with empty vector (EV) or *TP53INP1* plasmid. **B.** Cell cycle profile of cells transfected like in like in A, and treated with HU. **C.** WB indicating the levels of p-ATR, and *TP53INP1* in control HCT116 cells (EV) or overexpressing *TP53INP1*. Cells are not treated or treated with CTP or HU as indicated. **D.** Expression levels of *IncREST* and *TP53INP1* in untreated and HU-treated cells where *IncREST* is depleted with LNA (*IncREST*-KD), *TP53INP1* is depleted with siRNA (*TP53INP1*-KD) or both (Double-KD). **E.** Cell cycle profile of cells depleted of *TP53INP1* and treated as indicated.

3) The protein interaction data (including NCL) is strong and interesting but it would be important to determine whether other lncRNAs identified in Fig. 1 as S-phase/replication fork-specific and chromatin-bound are also enriched for these factors. It is conceivable that this might reflect a general protein/nascent RNA aggregation feature in the chromatin at stalled replication forks analogous to SAF-A (Creamer et al Mol Cell 2021.)

This is an interesting idea that we have to consider, since we don't fully understand the mechanism underlying the contribution of *IncREST*-NCL interaction to the fork regulation, and cannot exclude that additional ncRNAs with similar or redundant functions are involved. To investigate this we have performed RT-qPCR on NCL RIP in non-treated and HU-treated HCT116 on a panel of top candidate lncRNAs overexpressed in the chromatin fraction of HCT116 cell treated with HU. We chose the top activated lncRNAs in the RNA-seq shown in Fig. 1C, making sure to include several candidates that are regulated by TFs involved in the replication stress response (Fig 1G). Overall, we don't observe a general association of these lncRNAs with NCL. In the HU-treated condition we could identify some candidates with some association with NCL. However, the level of enrichment is low compared to the observed for *IncREST* (Fig. R33)

Fig. R33: RNA immunoprecipitation with NCL antibody followed by RT-qPCR detection in non-treated (NT) or HU treated (HU) HCT116 cells of the top lncRNAs induced by RS.

BIBLIOGRAPHY

An, J., Huang, Y.C., Xu, Q.Z., Zhou, L.J., Shang, Z.F., Huang, B., Wang, Y., Liu, X.D., Wu, D.C., and Zhou, P.K. (2010). DNA-PKcs plays a dominant role in the regulation of H2AX phosphorylation in response to DNA damage and cell cycle progression. *BMC Mol Biol* 11, 18.

Chen, X., Sun, Y., Zhang, T., Shu, L., Roepstorff, P., and Yang, F. (2021). Quantitative Proteomics Using Isobaric Labeling: A Practical Guide. *Genomics Proteomics Bioinformatics* 19, 689-706.
Chmielewska, M., Dedukh, D., Haczkiwicz, K., Rozenblut-Koscisty, B., Kazmierczak, M.,

Kolenda, K., Serwa, E., Pietras-Lebioda, A., Krasikova, A., and Ogielska, M. (2018). The programmed DNA elimination and formation of micronuclei in germ line cells of the natural hybridogenetic water frog *Pelophylax esculentus*. *Sci Rep* 8, 7870.

Drews, R.M., Hernando, B., Tarabichi, M., Haase, K., Lesluyes, T., Smith, P.S., Morrill Gavarro, L., Couturier, D.L., Liu, L., Schneider, M., et al. (2022). A pan-cancer compendium of chromosomal instability. *Nature* 606, 976-983.

Goder, A., Emmerich, C., Nikolova, T., Kiweler, N., Schreiber, M., Kuhl, T., Imhof, D., Christmann, M., Heinzl, T., Schneider, G., *et al.* (2018). HDAC1 and HDAC2 integrate checkpoint kinase phosphorylation and cell fate through the phosphatase-2A subunit PR130. *Nat Commun* 9, 764.

Hao, Q., Zong, X., Sun, Q., Lin, Y.C., Song, Y.J., Hashemikhabir, S., Hsu, R.Y., Kamran, M., Chaudhary, R., Tripathi, V., *et al.* (2020). The S-phase-induced lncRNA SUNO1 promotes cell proliferation by controlling YAP1/Hippo signaling pathway. *Elife* 9.

Ichijima, Y., Sakasai, R., Okita, N., Asahina, K., Mizutani, S., and Teraoka, H. (2005). Phosphorylation of histone H2AX at M phase in human cells without DNA damage response. *Biochem Biophys Res Commun* 336, 807-812.

John, R., Mathew, J., Mathew, A., Aravindakumar, C.T., and Aravind, U.K. (2021). Probing the Role of Cu(II) Ions on Protein Aggregation Using Two Model Proteins. *ACS Omega* 6, 35559-35571.

Ma, N., Matsunaga, S., Takata, H., Ono-Maniwa, R., Uchiyama, S., and Fukui, K. (2007). Nucleolin functions in nucleolus formation and chromosome congression. *J Cell Sci* 120, 2091-2105.

Mazumder, S., DuPree, E.L., and Almasan, A. (2004). A dual role of cyclin E in cell proliferation and apoptosis may provide a target for cancer therapy. *Curr Cancer Drug Targets* 4, 65-75.

Nakatani, T., Lin, J., Ji, F., Ettinger, A., Pontabry, J., Tokoro, M., Altamirano-Pacheco, L., Fiorentino, J., Mahammadov, E., Hatano, Y., *et al.* (2022). DNA replication fork speed underlies cell fate changes and promotes reprogramming. *Nat Genet* 54, 318-327.

Seillier, M., Peugeot, S., Gayet, O., Gauthier, C., N'Guessan, P., Monte, M., Carrier, A., Iovanna, J.L., and Dusetti, N.J. (2012). TP53INP1, a tumor suppressor, interacts with LC3 and ATG8-family proteins through the LC3-interacting region (LIR) and promotes autophagy-dependent cell death. *Cell Death Differ* 19, 1525-1535.

Shechter, D., Costanzo, V., and Gautier, J. (2004). ATR and ATM regulate the timing of DNA replication origin firing. *Nat Cell Biol* 6, 648-655.

Sirbu, B.M., McDonald, W.H., Dungrawala, H., Badu-Nkansah, A., Kavanaugh, G.M., Chen, Y., Tabb, D.L., and Cortez, D. (2013). Identification of proteins at active, stalled, and collapsed replication forks using isolation of proteins on nascent DNA (iPOND) coupled with mass spectrometry. *J Biol Chem* 288, 31458-31467.

Snyder, A.R., Zhou, J., Deng, Z., and Lieberman, P.M. (2009). Therapeutic doses of hydroxyurea cause telomere dysfunction and reduce TRF2 binding to telomeres. *Cancer Biol Ther* 8, 1136-1145.

Wang, L., Li, J., Zhou, H., Zhang, W., Gao, J., and Zheng, P. (2021). A novel lncRNA Discn fine-tunes replication protein A (RPA) availability to promote genomic stability. *Nat Commun* 12, 5572.

Wilhelm, T., Magdalou, I., Barascu, A., Techer, H., Debatisse, M., and Lopez, B.S. (2014). Spontaneous slow replication fork progression elicits mitosis alterations in homologous recombination-deficient mammalian cells. *Proc Natl Acad Sci U S A* *111*, 763-768.

Ye, C.J., Sharpe, Z., Alemara, S., Mackenzie, S., Liu, G., Abdallah, B., Horne, S., Regan, S., and Heng, H.H. (2019). Micronuclei and Genome Chaos: Changing the System Inheritance. *Genes (Basel)* *10*.

Yildirim, O., Izgu, E.C., Damle, M., Chalei, V., Ji, F., Sadreyev, R.I., Szostak, J.W., and Kingston, R.E. (2020). S-phase Enriched Non-coding RNAs Regulate Gene Expression and Cell Cycle Progression. *Cell Rep* *31*, 107629.

REVIEWERS' COMMENTS

Reviewer #1 (Remarks to the Author):

In this revision, Statello et al have performed new experiments and added new data to the manuscript. They addressed most of my comments and revised the text accordingly. The revisions are appropriate but I have a few comments that the authors may consider addressing with the text revision. In summary, I believe that this manuscript will be of broad interest to the lncRNA and cell cycle community.

- Although RIP and in vitro pulldown were used to validate NCL binding to lncREST, RIP is known to lead to re-association of RBP post-lysis (see PMID: 15388877) and is also known that RIP cannot isolate the specific binding sites. CLIP should have been used instead. Did the authors maybe examine iCLIP NCL data published in the literature?
- I agree that TMT and SILAC methods don't provide the same sensitivity as the label free methods. Nevertheless, label free methods can still be quantitative since without quantification it is difficult to define what is the background in these pulldown experiments. Thus, the volcano plot and p-values should be possible to generate. Please see PMID: 26496610 where the authors used the label free methods and provided volcano plots and p values (fig S4).
- Replication stress is usually induced by low concentration of APhiI such as 0.2uM (200nM) or 0.4 uM (400nM) for 24hr (PMID: 36266663, PMID: 31395887). Why the concentration used in the manuscript was 10 times higher?
- Reduction in markers of RS in lncREST KD cells upon HU (the new Fig 3E) should be quantified over controls from this representative experiment.
- How the authors explained that they did not observe major changes in cell cycle progression (new fig R1) in untreated lncREST KD cells while in the fig 7C depletion of lncREST leads to strong proliferation defects (untreated cells only). Could it be the timing?

Reviewer #2 (Remarks to the Author):

The authors have done an outstanding job in addressing my previous concerns. These new data now strengthen the claims and conclusions of the manuscript and I recommend publication of this study in Nature Communications.

Reviewer #3 (Remarks to the Author):

Statello et al have considered the reviewers' comments carefully and have been highly responsive. Thus, the revised manuscript represents a significant body of work and is appropriate for publication in Nature Communications.

My final request is that the authors include Figure R32 in the final version of the manuscript. While the data are clearly complex, including the observation that TP53INP1 knockdown and overexpression both lead to lncREST downregulation, the experiments have been carefully performed and add to our understanding of lncREST biology. In text/discussion, the interplay between TP53INP1 and lncREST can be described as complex and requiring further investigation.

REVIEWERS' COMMENTS

Reviewer #1 (Remarks to the Author):

In this revision, Statello et al have performed new experiments and added new data to the manuscript. They addressed most of my comments and revised the text accordingly. The revisions are appropriate but I have a few comments that the authors may consider addressing with the text revision. In summary, I believe that this manuscript will be of broad interest to the lncRNA and cell cycle community.

- Although RIP and in vitro pulldown were used to validate NCL binding to lncREST, RIP is known to lead to re-association of RBP post-lysis (see PMID: 15388877) and is also known that RIP cannot isolate the specific binding sites. CLIP should have been used instead. Did the authors maybe examine iCLIP NCL data published in the literature?

We are aware that re-association of proteins post-lysis can occur in RIP experiments, however the experimental settings used in the publication PMID: 15388877 use different experimental conditions than ours. The authors claim that whether proteins re-associate post lysis, as well as the extent of such re-association, depend not only on intrinsic factors, but also on the experimental conditions used: while they test their hypotheses using a native protocol, we performed UV-RIP, where the interactions RNA:proteins taking place *in vivo*, are stabilized covalently by crosslinking with UV before cell collection. Moreover, in the same paper the authors mention that a similar experimental approach was used to identify NCL binding proteins and no such re-association was observed in this case (quoting: “*For instance, a similar experimental approach indicated that ribonucleoprotein complexes associated with the RNA-binding protein nucleolin do not undergo significant rearrangements after cell lysis*”). We consider that this statement supports the legitimacy of our data suggesting a direct interaction between NCL and lncREST. Even if we agree with the reviewer that iCLIP is the best way to test NCL direct interactors, we used different orthogonal approaches to confirm the interaction between lncREST and NCL, which all led to the same conclusions. Indeed, UV-RIP was used to confirm the results of a lncREST pulldown (Figure 6A), and we further confirmed this interaction by *in vitro* pulldown (Figure 6B), which allowed us to identify the region of lncREST interacting with NCL. Therefore, we believe that our data are valid. Following the reviewer’s suggestion, we searched for iCLIP data for NCL, however, we could not find available iCLIP data on human cell lines, so we could not check for the presence of lncREST among the interacting RNAs, since lncREST is not conserved.

- I agree that TMT and SILAC methods don’t provide the same sensitivity as the label free methods. Nevertheless, label free methods can still be quantitative since without quantification it is difficult to define what is the background in these pulldown experiments. Thus, the volcano plot and p-values should be possible to generate. Please see PMID: 26496610 where the authors used the label free methods and provided volcano plots and p values (fig S4).

We thank the reviewer for this suggestion. In principle it is possible to generate a volcano plot with label free methods, however in our case it is not applicable. This type of representation is performed by plotting the ratio heavy/light for peptides, coming from quantitative proteomics comparing two different conditions. Unfortunately, our data are qualitative, not quantitative. Moreover, since we only have one LacZ sample, it is not possible to retrieve a meaningful p value. Even if we agree that quantitative methods are preferred for this type of analysis, a similar approach as ours is generally recognized and has been

widely applied for the identification of proteins associating to lncRNAs (e.g. PMID: 34321241, PMID: 34552092, PMID: 32858747)

- Replication stress is usually induced by low concentration of APHI such as 0.2uM (200nM) or 0.4 uM (400nM) for 24hr (PMID: 36266663, PMID: 31395887). Why the concentration used in the manuscript was 10 times higher?

We have taken the reviewer's suggestion in the first revision to perform key experiments with aphidicolin to strengthen our results. The reviewer has suggested to use 400uM (quoting the reviewer's question: *To confirm the role of IncREST in RS, the authors should use another drug used to activate RS such as aphidoclin (APHI, low dose 400uM, 24hr). Use of APHI will strengthen their data that IncREST is upregulated by RS and not caused by indirect DNA damage.*) Knowing that much lower doses of APH are able to induce replication stress, we have used a concentration that is in the middle between the lowest effective concentrations known in literature and the concentration requested by the reviewer.

- Reduction in markers of RS in IncREST KD cells upon HU (the new Fig 3E) should be quantified over controls from this representative experiment.

We have added the quantification of the WB shown in Figure 3E.

- How the authors explained that they did not observe major changes in cell cycle progression (new fig R1) in untreated IncREST KD cells while in the fig 7C depletion of IncREST leads to strong proliferation defects (untreated cells only). Could it be the timing?

We thank the reviewer for this question. Indeed, we did not observe a significant change in cell cycle progression in NT samples, even if the proliferation of IncREST depleted cells is impaired, as we show in Fig. 7C. We don't believe the impaired proliferation is linked to cell cycle defects in this case, but rather to increased apoptosis of these cells, which we observe already in NT condition, as shown in Supplementary Fig. 7C. Apoptosis can be caused by different factors in these cells. One major factor could be the alterations of NCL distribution in IncREST depleted cells. It has previously been shown that impairment of NCL expression leads to altered proliferation and ultimately apoptosis both in vivo and in vitro through several mechanisms, including the PI3K/AKT pathway, or through accumulation of mitotic defects, such as those that we have also observed in IncREST depleted cells (see Figure 7 and Supplementary Figure 7), which recapitulate NCL knockdown (PMID: 35128835, PMID: 17535846, PMID: 17692122).

Reviewer #2 (Remarks to the Author):

The authors have done an outstanding job in addressing my previous concerns. These new data now strengthen the claims and conclusions of the manuscript and I recommend publication of this study in Nature Communications.

We thank the reviewer for the positive feedback on our revision work.

Reviewer #3 (Remarks to the Author):

Statello et al have considered the reviewers' comments carefully and have been highly responsive. Thus, the revised manuscript represents a significant body of work and is appropriate for publication in Nature Communications.

My final request is that the authors include Figure R32 in the final version of the manuscript. While the data are clearly complex, including the observation that TP53INP1 knockdown and overexpression both lead to IncREST downregulation, the experiments have been carefully performed and add to our understanding of IncREST biology. In text/discussion, the interplay between TP53INP1 and IncREST can be described as complex and requiring further investigation.

We thank the reviewer for the positive comments. We have now added the data shown in Figure R32 of our previous rebuttal to Supplementary Figure 3, as panels G, H, K, and L-N, and discussed them in the main text of the manuscript.